# Enhancing multiclass COVID-19 prediction with ESN-MDFS: Extreme smart network using mean dropout feature selection technique

**Saghir Ahmed[1], Basit Raza[1]\*, Lal Hussain[2,3]\*, Touseef Sadiq[4]\*, Ashit Kumar Dutta[5]**

**1** Department of Computer Science, COMSATS University, Islamabad Capital Territory, Islamabad, Pakistan, **2** Department of Computer Science & IT, Neelum Campus, The University of Azad Jammu and Kashmir, Athmuqam, Azad Kashmir, Pakistan, **3** Department of Computer Science & IT, King Abdullah Campus, The University of Azad Jammu and Kashmir, Muzaffarabad, Azad Kashmir, Pakistan, **4** Department of Information and Communication Technology, Centre for Artificial Intelligence Research (CAIR), University of Agder, Grimstad, Norway, **5** Department of Computer Science and Information Systems, College of Applied Sciences, AlMaarefa University, Ad Diriyah, Riyadh, Kingdom of Saudi Arabia

\* touseef.sadiq@uia.no (TS); basit.raza@comsats.edu.pk (BR); lall_hussain2008@live.com (LH)

**Data Availability Statement:** "The datasets used in this study are publicly available. The COVID-19 chest X-ray (CXR) images were obtained from

## Abstract

Deep learning and artificial intelligence offer promising tools for improving the accuracy and efficiency of diagnosing various lung conditions using portable chest x-rays (CXRs). This study explores this potential by leveraging a large dataset containing over 6,000 CXR images from publicly available sources. These images encompass COVID-19 cases, normal cases, and patients with viral or bacterial pneumonia. The research proposes a novel approach called "Enhancing COVID Prediction with ESN-MDFS" that utilizes a combination of an Extreme Smart Network (ESN) and a Mean Dropout Feature Selection Technique (MDFS). This study aimed to enhance multi-class lung condition detection in portable chest X-rays by combining static texture features with dynamic deep learning features extracted from a pre-trained VGG-16 model. To optimize performance, preprocessing, data imbalance, and hyperparameter tuning were meticulously addressed. The proposed ESN-MDFS model achieved a peak accuracy of 96.18% with an AUC of 1.00 in a six-fold cross-validation. Our findings demonstrate the model's superior ability to differentiate between COVID-19, bacterial pneumonia, viral pneumonia, and normal conditions, promising significant advancements in diagnostic accuracy and efficiency.

## 1. Introduction

COVID-19 pandemic is the greatest incident affecting the lives of many people and thus a major issue concerned is the need to have an accurate diagnosis. However, [1] researchers stress that diagnosing the disease can be done with the help of the Real-Time reverse transcription-Polymerase Chain Reaction (RT-PCR) method, whereas some researchers [2] indicate that biomarkers are of great value during the early ages of the disease diagnosis. The pandemic has also been responsible for a shift in the mortality rates, with older people being the ones

Cohen et al. via GitHub (https://github.com/ieee8023/covid-chestxray-dataset). Additional CXR images were sourced from Radiopaedia (https://radiopaedia.org/), The Cancer Imaging Archive (TCIA) (https://www.cancerimagingarchive.net/), and SIRM (https://www.sirm.org/category/senza-categoria/covid-19/ & https://sirm.org/?s=COVID-19). The pneumonia CXR images (N = 3863) and normal (healthy) CXR images were acquired from the Kaggle repository (https://www.kaggle.com/paultimothymooney/chestxray)."

**Funding:** Ashit Kumar Dutta would like to express sincere gratitude to AlMaarefa University, Riyadh, Saudi Arabia, for providing funding to conduct this research.

**Competing interests:** The authors have declared that no competing interests exist

**Abbreviations:** CXRs, chest x-rays; ESN-MDFS, Extreme Smart Network using Mean Dropout Feature Selection Technique; RT-PCR, Real-Time reverse transcription-Polymerase Chain Reaction; CT, Computed Tomography; CNNs, Convolutional Neural Networks; GAN, Generative Adversarial Networks; DL, Deep learning; GLCM, Grey Level Co-occurrence Matrix; CLs, convolutional layers; FLs, fully connected layers; ILSVRC, ImageNet Large Scale Visual Recognition Challenge (); VGG, Visual Geometry Group (); ReLU, Rectified Linear Unit; ROC, Receiver Operating characteristic; AUC, Area Under the Curve.

who have mostly died [3,4] draws our attention to various comorbidities and organ injuries which transform the treatment plan of COVID-19 in the clinical setting. These research, by virtue of the fact, stress the existence and severity of dire results associated with delayed and improper diagnosis which can be a real threat to the survival of humankind. The latest studies have revealed that deep learning models can be used for coronavirus image classification. The researchers [5] managed to achieve an accurate detection COVID-19 from X-ray images with ResNet101. The researchers of [6] also reported a high accuracy using the CNN. In the paper authored by [7], the same Xception model was combined with an additional channel attention mechanism for the automated classifier of the Computed Tomography (CT) scan of COVID-19 case making it extremely precise. The studies taken together imply that the deep learning models can identify COVID-19 cases through images, which is useful in medical diagnosis and disease detection. It is said that the sophisticated and palatial deep learning model is much greater than its thinner versions. This is because the deeper complex models and algorithms that are sized bigger have their own advantages and disadvantages. From one point of view, the large size of deep learning models allows them to capture more complicated and subtle patterns in the data, thus, improving their performance and accuracy [8].

While deep learning models have shown promise in classifying COVID-19 images, there are still limitations that need to be addressed. One such limitation is the lack of research on applying domain adaptation techniques to overcome the challenge of the cross-dataset problem. Existing solutions, such as COVID19-DANet, have shown some promise but still require further improvement to achieve better results across different datasets [9]. Deep learning models, including Convolutional Neural Networks (CNNs), perform well when trained and tested on the same dataset but show significantly lower performance when applied to different datasets. This indicates a lack of generalization across different data sources. The problems of COVID-19 classification, also problem that the quality and quantity of available COVID-19 image datasets vary significantly, which affects the training and performance of deep learning models. Inconsistent data quality can lead to unreliable model predictions [10]. Data enhancement methods are crucial for improving model performance, but there is a need for more sophisticated techniques to handle the variability in COVID-19 image datasets effectively. Supervised learning methods, while effective, require large amounts of labelled data, which is often scarce in the context of COVID-19. The researchers [11] highlights the challenges of extensive computational resources, limited annotated datasets, and a large amount of unlabeled data. The researchers [12] further emphasizes the difficulty of assessing severity due to small datasets and the high dimensionality of images. The authors [13,14] both note the limitations of single task learning and the need for more efficient models. These studies collectively underscore the need for more robust and efficient deep learning models in COVID-19 image classification. These limitations hamper the development of robust models. Semi-supervised and unsupervised learning methods have been explored, but they still face challenges in achieving high accuracy and reliability in COVID-19 image classification.

The authors [15]proposed a novel automated framework for the classification of tuberculosis, COVID-19, and pneumonia from chest x-ray images using deep learning and an improved optimization technique. The proposed deep learning-based framework achieved high classification accuracy 98.2%, 99.0%, and 98.7%) on three different datasets for tuberculosis, COVID-19, and pneumonia detection from chest X-ray images. The authors employed the Wilcoxon signed-rank test to statistically validate the superior performance of their proposed method. The integration of feature fusion was instrumental in enhancing the method's accuracy. The researchers [16] proposed a wrapper-based technique to improve the classification performance of chest infection (including COVID-19) detection using X-rays by extracting deep features using pretrained deep learning models and optimizing them using various optimization

techniques, while also using a network selection technique to select the deep learning models. The proposed deep learning framework achieved a high classification accuracy of 97.7% in detecting chest infections, including COVID-19. Rigorous validation confirmed the framework's reliability for classifying both COVID-19 and other chest infections, suggesting its potential as a valuable tool for clinicians.

If we further look into the problems of COVID-19, the one major constraint is the reliance on binary classifiers or building classifiers based on only a few classes, hindering comprehensive classification [17]. Additionally, most studies focus on flat single-feature imaging modalities without incorporating clinical information or utilizing the hierarchical structure of pneumonia, leading to clinical challenges [18]. The availability of limited COVID-19 imaging data poses a challenge for developing effective automated picture segmentation methods, impacting quantitative assessment and disease monitoring [19]. Moreover, the biggest challenge lies in the availability of training data, with data augmentation methods like Generative Adversarial Networks (GAN) -based augmentation found to be subpar compared to classical methods for COVID-19 image classification [19,20].

This paper explores how the size of a deep learning model influences different tasks and identifies certain implications when working with such models. Larger models have been seen to be associated with higher computational costs and more parameters especially when dealing with complex models meaning that the models cannot be easily deployed in any device with limited resource capabilities [21]. However, prior research has demonstrated that such approaches may not always be the case, such as in the application of CNNs to detect brain tumours where less input sizes describe heightened accuracy coupled with enhanced training times, evidenced from the use of the 64px inputs models as opposed to the larger input models [22]. However, in practice, when using models such as autoencoders in deep learning on datasets where features are globally similar but locally dissimilar, the authors noted that using smaller batch sizes improve model performance and yield better biologically relevant information, which raises the cost of batch size when design the model [23]. The measures regarding challenges of large deep learning (DL) models consists of methods like degree of parallelism, lighter data matrices and system enhancement to increase the potential [24].

Lightweight deep learning models have the capability to reduce the model size and memory requirements, and to optimize the model in order to make it efficient to implement on edge device. The primary goal of these models is to minimize computational complexity while still delivering high performance [25,26]. For example, a lightweight deep learning model created for identifying human posture exhibited a significantly smaller size of 46.2 MB, in contrast to the baseline model's 227.8 MB [27]. Similarly, a model developed for detecting ophthalmic diseases was reported to be ten times lighter than the popular biomedical segmentation model UNet, with a memory size of around 35 MB. These smaller models play a critical role in facilitating real-time processing on battery-operated devices and enable efficient deployment on edge devices with limited resources.

The advancement of lightweight CNN models proves beneficial for diverse applications, particularly in situations with constrained computational resources. These models offer advantages such as reduced inference delay, minimal memory requirements for deployment on embedded devices, and the ability to swiftly update over-the-air [28]. For deployment on mobile devices with limited resources, lightweight Convolutional Neural Network (CNN) models are crucial. These models achieve high accuracy while keeping computational costs low. MobileNetV2 model utilizes depth-wise separable convolutions and inverted residual connections to reduce computations without sacrificing accuracy. It achieves state-of-the-art performance on various tasks, making it ideal for mobile vision applications [29]. ShuffleNet is another lightweight architecture specifically designed for mobile devices with limited

computational power. It employs pointwise group convolutions and channel shuffle operations to achieve lower computational demands while maintaining accuracy [30]. EfficientNet is from the family of models surpasses previous CNNs in both accuracy and efficiency. Obtained by scaling up MobileNets and ResNets using neural architecture search, EfficientNets achieve state-of-the-art accuracy on diverse datasets while being smaller and faster during inference compared to other models [31]. Furthermore, techniques such as pruning, quantization, and knowledge distillation can further reduce the size of CNN models. This makes them even more suitable for deployment on resource-constrained devices [32].

Lightweight CNN models are a game-changer for real-time video surveillance. They offer several key advantages: minimal inference delay, meaning they process video frames quickly for real-time analysis, low memory requirements allowing them to run on resource-constrained devices, and the capability to be trained, fine-tuned, and deployed in a distributed manner [28]. This makes them ideal for embedded systems and facilitates efficient processing of large video datasets for wider deployment. The emergence of lightweight deep learning methodologies has gained prominence due to their ability to facilitate efficient and real-time processing on edge devices. These methodologies can be broadly categorized into two approaches: developing lightweight deep learning algorithms from scratch and transforming existing models into more compact versions. Researchers have explored various lightweight models, such as SqueezeNet, ShuffleNet, and MobileNet, comparing their performance parameters with conventional models like AlexNet and GoogleNet [33]. These lightweight models have shown promising results across numerous daily life applications. Moreover, lightweight deep learning algorithms have been applied to studying slip performance in composite materials used in construction, showcasing their versatility [34]. Overall, lightweight deep learning techniques offer a promising avenue for efficient processing in resource-constrained environments, facilitating real-time processing and reducing computational complexity.

The size of trained models presents deep learning models for COVID-19 classification on edge devices hence important to design light models. Various studies have pointed out that, it is beneficial to exist in the method to decrease the number of model parameters while retaining high accuracy in order to optimize model implementation in the edge environment [35, 36, 37]. Such as attention modules and mixed loss functions has been suggested to reduce the size of models while incurring a similar level of performance so that the models can effectively be deployed on edge devices that have restricted resources [38]. While models such as MobileNetV2 are more lightweight, they have emerged dominant in performance with constrained memory needs to increase the efficacy of deploying the model within edge devices [39]. The efficient deep learning neural networks combined with wearable medical sensors can be embedded in smartphone applications and similar other devices, preserving patient privacy and ensuring efficient resource use[40]. Based on all above evidence we suggest that the size of trained models depends heavily on machine learning and AI implementation when deployed for COVID-19 classification on edge devices. They will remain pertinent due to the need for model compression, selective ensemble methods, and other developments like mixed-precision training. The idea is to allow the precise and real-time execution of deep learning models on source devices, which may help accelerate and improve the COVID-19 detection.

Distinguishing COVID-19 from other lung infections remains a challenging task. While researchers are actively developing tools to improve prediction performance, limitations persist in the preprocessing and processing stages. This study addresses these challenges by focusing on the preprocessing stage. We propose a novel approach that utilizes median filtering and interpolation methods to remove noise from the imaging data. Additionally, we address data imbalance using data augmentation techniques and a stratified 5-fold cross-validation strategy to prevent overfitting and ensure a balanced distribution for training and validation purposes.

Furthermore, we optimize the hyperparameters of the VGG-16 deep learning algorithm through a grid search method. Finally, we introduce the ESN-MDFS system, a novel approach that combines an Extreme Smart Network (ESN) with a Mean Dropout Feature Selection Technique (MDFS). This system aims to improve multi-class detection (COVID-19, normal, viral pneumonia, and bacterial pneumonia) by extracting static features using Grey Level Co-occurrence Matrix (GLCM) analysis and dynamic features through the pre-trained VGG-16 model.

## 2. Materials and methods

### 2.1. Proposed model

This study enhances multiclass COVID-19 prediction through a novel approach encompassing the following key elements as reflected in Fig 1A and 1B:

- **Optimized pre-processing:** Chest X-ray image quality was improved using techniques such as interpolation, data cleaning, augmentation, feature engineering, image enhancement, morphological operations, segmentation, and transformation.

- **Feature extraction:** Dynamic VGG-19 and static GLCM features were computed from multiclass data to capture diverse image characteristics.

- **Feature selection:** A hybrid feature space (HFS) was refined using feature selection methods to eliminate redundant features, thereby improving prediction performance and model size for efficient deployment on edge devices

- The optimal HFS was then utilized to the robust optimized XGBoost algorithm for improved prediction

- **Hyperparameter tuning:** The hyperparameters of the XGBoost machine learning algorithm were meticulously optimized.

 **Deep features extracted from VGG-19** provide a powerful representation of image content. They capture high-level semantic information about the image, such as the presence of specific objects or patterns. In the context of COVID-19 classification, these features can effectively discriminate between different lung pathologies, including pneumonia, viral pneumonia, and COVID-19. By leveraging the hierarchical structure of VGG-19, these features can capture subtle visual patterns that are often challenging for traditional image processing techniques.

 **Static GLCM features,** on the other hand, provide complementary information about the texture and spatial relationships between pixels in an image. These features are sensitive to image patterns and structures, which can be crucial for differentiating between different types of lung abnormalities. By combining deep features and GLCM features, it is possible to create a more robust and discriminative feature space for multi-class COVID-19 classification.

### The hybrid feature space (HFS)

- Deep features and GLCM features capture different aspects of image information, leading to improved classification performance.

- The combination of these features can better differentiate between subtle visual patterns associated with different lung diseases.

- The use of multiple feature types can help to reduce the impact of noise and variations in image quality.

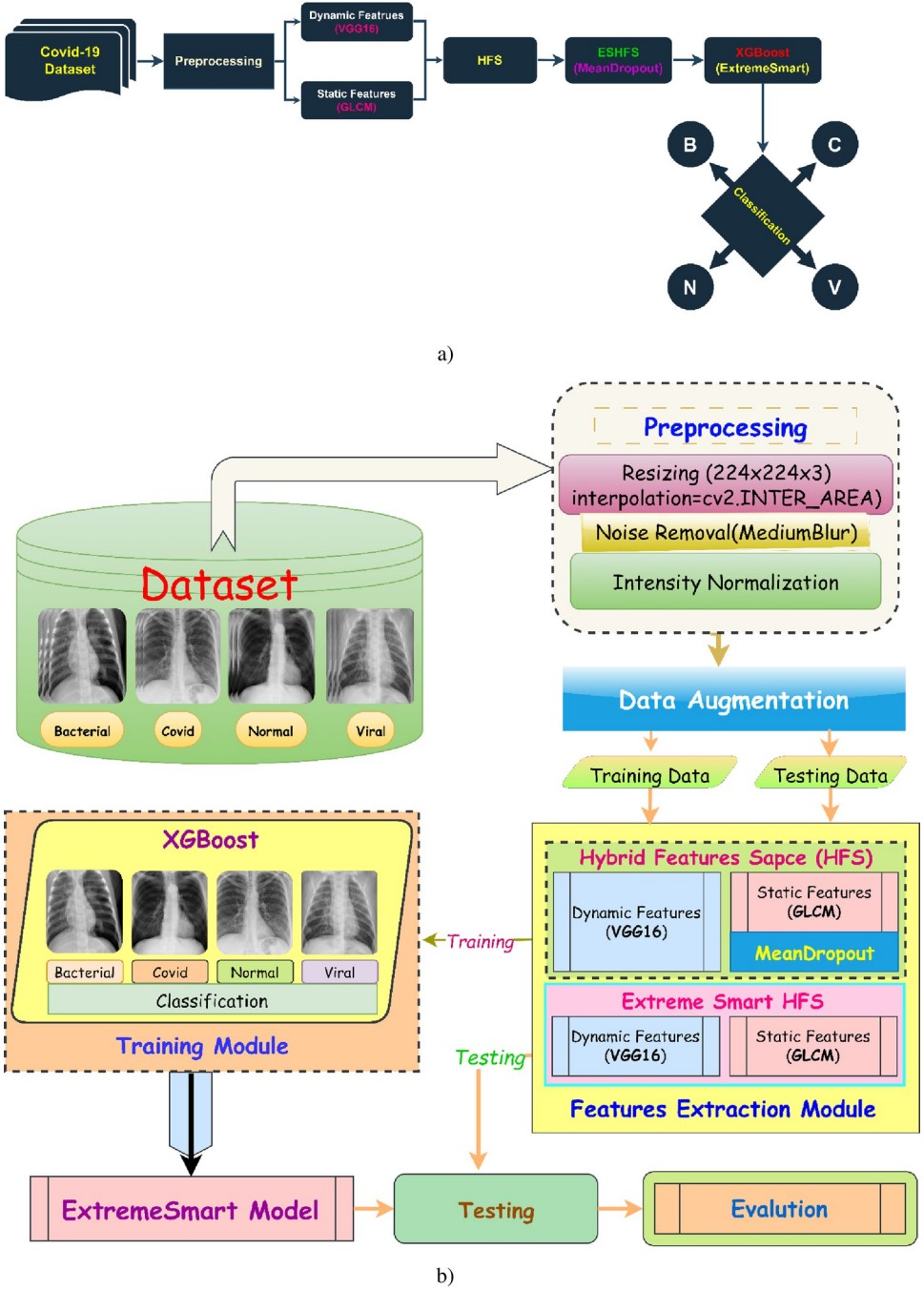

**Fig 1.** The proposed Enhancing Multiclass COVID Prediction with ESN-MDFS: Extreme Smart Network using Mean Dropout Feature Selection Technique Diagram (a) shows the entire workflow of the proposed technique {B: Bacterial, C: Covid-19, N: Normal and V: Viral}, whereas diagram (b) shows the phases of the proposed technique in detail.

By effectively fusing these features and employing appropriate machine learning techniques, we developed highly accurate and reliable COVID-19 classification models.

## 2.2. Proposed model algorithm: Enhancing COVID Prediction with ESN-MDFS

- *Preprocessing Step*:
  *foreach image in imageDataset*

  > *Apply interpolation during image resizing (224,224)*

  > *Apply medianBlur for denoising*

  > *Apply intensity Normalization*

- *end foreach*

- *Data Augmentation*:
  *foreach class in classes*

  > *find difference in classes*

  > *apply data augmentation using library imageDataGenerator*

- *end foreach*

- *Data Split*
  *Split dataset into train and test using train_test_split method*

  > *train = 0.8*

  > *test = 0.2*

- *Features Selection*:

  > *Static Features–using GLCM (25 Features)*

  > *Dynamic Features–using VGG16 (1024 Features)*

- *Hybrid Features Space*:

  > *Combined Static and Dynamic Features–HFS*

  > *Apply Mean Dropout Technique for Selection of Important Features from HFS*

- *Train Models on Train and Test*

  > *Train XGBoost Model on HFS*

  > *generating ROC, Confusion_matrix, Classification_Report*

- *Deploy Model*

  > *Deploy smart model on edge devices*

### 2.3. Dataset

To train our deep CNN for distinguishing COVID-19 from other pneumonia types, we leveraged a diverse dataset compiled from several publicly available sources, similar to the approach used in previous studies [41–43]. This dataset incorporates chest X-ray images of COVID-19 (N = 1525): sourced from Cohen et al. via GitHub [44], Radiopaedia, SIRM, TCIA, and Pneumonia (N = 3863): retrieved from the Kaggle repository, Normal individuals (N = 1525): sourced from the Kaggle repository and the NIH dataset. This multifaceted dataset,

encompassing images from various public sources, strengthens the generalizability and robustness of our model.

## 2.4. Preprocessing

**2.4.1. Image preprocessing.** To unlock valuable insights from images, we employ image preprocessing. This vital step refines image quality and readies it for further analysis. Fig 2 showcases three key aspects of this transformation: noise reduction for a clearer view, feature enhancement for sharper details, and normalization for seamless integration into subsequent steps.

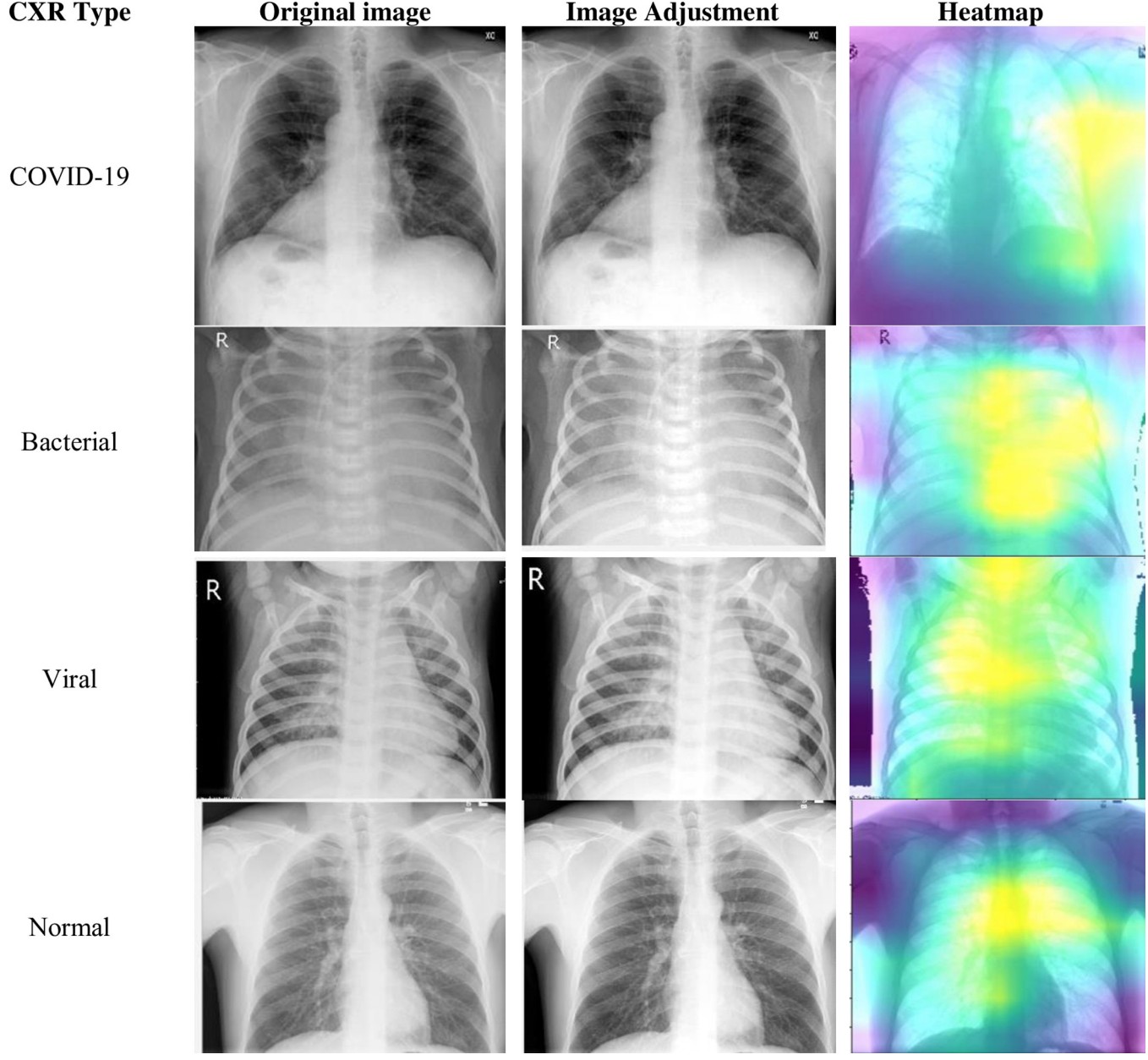

**Fig 2. Image pre-processing steps.**

**1. Interpolation:**

- Interpolation deals with the problem of missing data in an image. It fills in the gaps in the image data with estimated values based on the surrounding pixels. This can be important for tasks such as image resizing, scaling, and registration.

**2. Noise Removal:**

- Noise is unwanted information that can corrupt an image and interfere with subsequent analysis. It can originate from various sources, such as sensor imperfections, environmental factors, and data transmission errors.

- Noise removal aims to remove or suppress this noise while preserving the true image content. Various noise removal filters are available, each targeting specific types of noise. Common filters include median blur filter.

**3. Intensity Normalization:**

- Intensity normalization aims to adjust the intensity values of an image to a desired range. This can be necessary for tasks such as image registration, segmentation, and feature extraction.

- We utilized histogram equalization.

**2.4.2. Data augmentation.** To compensate for potential limitations in the dataset, we employed two crucial strategies: data augmentation and stratified splitting. Data augmentation artificially expands the dataset by generating variations of existing samples, making the model more robust to real-world variations and preventing overfitting [45]. Techniques like adding noise, applying transformations, and generating synthetic data were utilized to achieve equal representation of all classes, further enhanced by stratified splitting during data division. This ensures each training and test set accurately reflects the distribution of classes in the original data. The Fig 3 depicts the data augmentation methods.

- **Rotation:**
  Randomly rotates the image by an angle within a predefined range. This can help the model learn to recognize objects from different angles. Range: -50 to 20 degrees, probability: 0.2.

- **Horizontal Flipping:**
  Flips the image horizontally (left-right). This can help the model learn to recognize objects that are not symmetrically aligned. **Probability:** 1.0

- **Vertical Flipping:**
  Flips the image vertically (top-bottom). This can help the model learn to recognize objects that are not symmetrically aligned. **Probability:** 1.0

- **Image Shearing:**
  Definition: Applies a shearing transformation to the image, distorting it in a parallelogram-like shape. This data augmentation technique helps the model develop viewpoint invariance, allowing it to recognize objects even when viewed from different angles. **Angle:** -40 to 40, **probability:** 0.2.

- **Gamma Contrast:**
  Adjusts the gamma value of the image, changing the overall brightness and contrast. This data augmentation technique can enhance the model's illumination invariance, allowing it to recognize objects even under varying lighting conditions. **Range:** 0.5 to 2, **pobability:** 0.2.

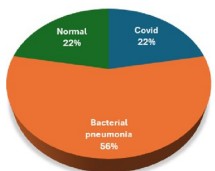
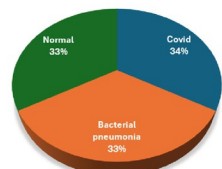
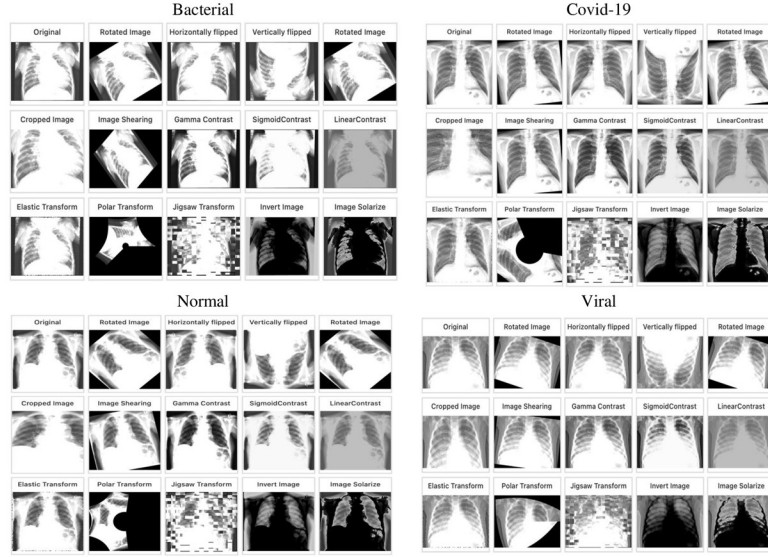

**Fig 3. Data Augmentation techniques on multiclass.**

- **Sigmoid Contrast:**
  Applies a sigmoid function to the image's pixel intensities, enhancing the contrast between dark and bright areas. This can help the model learn to recognize objects with different brightness levels. **Coefficient:** 5 to 10, **probability:** 0.2.

- **Linear Contrast:**
  Adds or subtracts a constant value to all pixel intensities of the image, adjusting the overall brightness. This can help the model learn to recognize objects under different lighting conditions. **Delta:** -0.2 to 0.2, **probability:** 0.2.

- **Elastic Transform:**
  Applies an elastic transformation to the image, distorting it in a rubber-like manner. This can help the model learn to recognize objects under different deformations. **Alpha:** 60, **Sigma:** 4, **Probability:** 0.2

- **Polar Transform:**
  Converts an image from Cartesian coordinates to polar coordinates, applies random rotations and shifts, and then converts back to Cartesian coordinates. This can help the model learn to recognize objects under different rotational perspectives. **Max magnitude: -0.2 to 0.7**, **probability:** 0.2.

- **Jigsaw Transform:**
  Divides an image into multiple smaller patches and randomly rearranges them, creating a new image. This can help the model learn to recognize objects from fragmented views.**grid size:** 4x4 to 8x8, **Pixel interpolation:** 3 to 7, **Probability:** 0.2.

- **Invert Image:**
  Negates all pixel intensities of the image, creating a negative image. This can help the model learn to recognize objects based on their shape and texture, not just their color. **Probability:** 1.0

- **Polarize Image:**
  Randomly increases or decreases the saturation of the image, creating a more intense or

muted color palette. This can help the model learn to recognize objects under different color conditions. **Factor:** 0.5 to 2, **Probability:** 0.2.

After the augmentation is applied the images are equalized to N = 2521.

**2.4.3. Image resize.** For image resizing, we opted for the "inter area" interpolation method, a technique often used in computer vision to estimate the value of a pixel based on its surroundings [46]. This specific method, utilizing the average values of nearby pixels, excels at producing smooth and accurate results, especially when downscaling images. Unlike some other interpolation methods, "inter area" considers the contributions of multiple surrounding pixels, leading to a visually pleasing and artifact-free outcome.

**2.4.4. Hyperparameters optimization.** Optimizing hyperparameters plays a crucial role in fine-tuning the performance of deep learning models like VGG-16. By adjusting these settings, we can achieve optimal accuracy, minimize overfitting, and improve the model's generalizability to unseen data.

Here's a breakdown of key VGG-16 hyperparameters and their potential grid search values:

1. Learning Rate: The learning rate governs the step size taken during gradient descent, affecting the speed of weight updates in the model. Grid Values: [0.0001, 0.001, 0.01, 0.1]

2. Momentum: Helps the model overcome shallow local minima by incorporating the direction of past gradients. Grid Values: [0.0, 0.5, 0.9, 0.99]

3. Weight Decay: Regularizes the model by penalizing large weights, preventing overfitting. Grid Values: [1e-4, 1e-5, 1e-6, 0]

4. Batch Size: Number of samples processed together during training. Grid Values: [8, 16, 32, 64.

5. Optimizer: Algorithm used to update the model's weights based on the loss function. Grid Values: [Adam, SGD, RMSprop]

6. Number of Training Epochs: Epochs (number of training dataset passes). Grid Values: [10, 20, 50, 100]

7. Activation Function: Embeds non-linear decision boundaries, empowering the network to capture intricate interactions between features. Grid Values: [ReLU, Leaky ReLU, tanh]

8. Dropout Rate: Randomly drops out neurons during training, preventing co-adaptation and improving generalizability. Grid Values: [0.2, 0.3, 0.4, 0.5]

9. Early Stopping: Monitors a validation metric and stops training when it stagnates, preventing overfitting. Grid Values: [Patience: 5, 10, 15]

**2.4.5. VGG-16 hyperparameter optimization.** To optimize the performance of our model, we employed a grid search method to identify the most effective hyperparameter settings. Following optimal values were selected as optimal: Learning Rate: 0.001, Momentum: 0.9, Weight Decay: 1e-5, Batch Size: 32, Optimizer: Adam, Epochs: 50, Activation: ReLU, Dropout: 0.3, Early Stopping Patience: 10.

## 2.5. Feature extraction

Deep features extracted from VGG-19 provide a powerful representation of image content. They capture high-level semantic information about the image, such as the presence of specific objects or patterns. In the context of COVID-19 classification, these features can effectively

discriminate between different lung pathologies, including pneumonia, viral pneumonia, and COVID-19. By leveraging the hierarchical structure of VGG-19, these features can capture subtle visual patterns that are often challenging for traditional image processing techniques.

Static GLCM features, on the other hand, provide complementary information about the texture and spatial relationships between pixels in an image. These features are sensitive to image patterns and structures, which can be crucial for differentiating between different types of lung abnormalities. By combining deep features and GLCM features, it is possible to create a more robust and discriminative feature space for multi-class COVID-19 classification.

**2.5.1. Static feature extraction based on Grey-level Co-occurrence Matrix (GLCM).** This approach utilizes GLCM, a texture feature extraction method introduced by Haralick in 1973, to analyze the input image [47].

The Gray Level Co-occurrence Matrix (GLCM) is a statistical technique used to extract texture features from images by analyzing the spatial relationships between pixel intensities. Its applications span various domains, including SAR imagery for land cover classification (water, vegetation, urban areas) [48] and medical imaging for detecting retinal abnormalities [49], where color features have shown superior accuracy. Gray Level Co-occurrence Matrix (GLCM) analysis computes the frequency of pixel pairs with specific intensity values and spatial relationships, forming a matrix from which statistical features can be extracted. In this study, 25 GLCM features were initially calculated and subsequently reduced to 17 through MeanDropout feature selection.

GLCM characterizes texture by analyzing the spatial relationships between neighboring pixels. This is achieved in two steps:

Step 1: Building the GLCM. Pixel pairs separated by a specific distance (d) and direction (θ) are counted and tabulated. This establishes a spatial relationship between a reference pixel and its neighbors.

Step 2: Feature extraction. From the GLCM, a set of scalar quantities is computed, each capturing different aspects of the original texture. These quantities, collectively forming the GLCM features, represent the frequency of various gray-level combinations occurring within the image [47].

The GLCM extracts texture features from images by analyzing the spatial relationships between pairs of pixels. Introduced in 1973 by Haralick et al. [34. GLCM characterizes texture through various statistical measures derived from the second-order statistics of the image. Obtaining GLCM features involves two steps:

1. **Spatial Co-occurrence Calculation**: For each pixel in the image, the frequency of its gray level co-occurring with the gray levels of its neighbors at a specific distance (d) and direction (θ) is tabulated. This establishes a spatial relationship between the reference pixel and its neighbors.

2. **Feature Extraction:** From the co-occurrence matrix, a set of scalar features are computed. These features capture various aspects of the original texture, such as contrast, homogeneity, and directionality.

The resulting GLCM matrix encodes the frequency of different gray level combinations within the image, providing valuable information about the underlying texture patterns [47]. Texture features computed from GLCM are Inverse Difference Moment [50], Contrast, Energy [50], Entropy [50], Cluster Shade, Sum of Average, Homogeneity [50–52], Sum of Square Variance [53], and Correlation [52] etc. The GLCM features are detailed and utilized in studies [53–56].

**2.5.2. Dynamic feature extraction from VGG-16 CNN model.** VGG-16 is a convolutional neural network (CNN) architecture that was proposed by researchers at the Visual Geometry Group (VGG) at the University of Oxford in 2014. It is named after the group that developed it and the fact that it has 16 weight layers (excluding the pooling and fully connected layers). The VGG-16 architecture was designed to participate in the ImageNet Large Scale Visual Recognition Challenge (ILSVRC) in 2014. The challenge involved classifying images into one of 1,000 categories. VGG-16, a convolutional neural network introduced by the Visual Geometry Group at Oxford University in 2014,achieved prominence through its success in the ImageNet Large Scale Visual Recognition Challenge. Its architecture, characterized by a series of small 3x3 convolutional filters, led to a model with 13 convolutional and 3 fully connected layers. While its depth enhanced feature extraction capabilities, it also increased susceptibility to overfitting on smaller datasets. Nevertheless, VGG-16 remains influential due to its strong performance on large-scale image classification tasks.

The VGG16 model was employed for feature extraction, generating 1024-dimensional feature vectors for each image in the dataset. To adapt the model to the specific characteristics of our target problem, the final four fully connected layers of VGG16 were re-trained on the selected datasets. Total trainable parameters for VGG-16 were 1,051,648 (first Dense) + 2,099,328 (second Dense) + 4,100 (final Dense) = 3,155,076 trainable parameters. To adapt the model to the specific task, the final four layers were fine-tuned using the selected datasets. Optimal performance hinges on careful selection of hyperparameters and we chosen the learning rate, optimizer, batch size, epochs, and regularization techniques by optimizing the hyperparameters using Bayesian optimization to fine-tune these hyperparameters.

**2.5.3. Hybrid feature model.** By fusing static GLCM features with dynamic features learned from the VGG-16 model, a robust hybrid feature space is created. This approach effectively leverages the complementary strengths of both modalities: GLCM's emphasis on textural information and VGG-16's extraction of high-level visual representations. The resulting feature space significantly enhances image analysis tasks, including classification, object recognition, and texture analysis.

**2.5.4. Mean dropout feature selection method for Hybrid Feature space (HFS).** We proposed a Mean dropout technique for feature selection, it define as follows:-

Let X be the set of data represented by the All_Features.

Let C be the last column index of the All_Feature, which contain the class of the features.

Let F be the set of features (columns other than the Class column).

Let Y be the set of unique class labels form the last column.

The equation of Mean Dropout represented as:

$$AllFeaturesMean(y, f) = \frac{1}{N_y} \sum_{x \in X} x[f] \cdot \big[ y = x[C] \big]$$

Where:

- *AllFeaturesMean*(*y*,*f*) represents the mean value of feature *f* with class *y*.

- $N_y$ is the number of instances in class y.

- *x*[*f*] represents the value of feature *f* for data point *x*

- [*y* = *x*[*C*]] is an indicator function that equals 1 if *y* is equal *x*[*C*], and 0 otherwise.

- The sum is taken overall datapoints $x$ in the dataset.

  In more concise manner we can represent above equation as below:

$$AllFeaturesMean(y) = \left\{ \frac{1}{N_y} \sum_{x \in X} x[f] \cdot [y = x[C]] \,|\, f \in F \right\}$$

Where:

- *AllFeaturesMean(y)* is a set of mean values for all feature within class $y$

- $f \,\epsilon\, F$ iterates over all features in the dataset.

  dropping all those features where mean values are different classes of same features is same, we can modify the above equation as follows:

$$SelectedFeatures\,(y) = f \in F \,|\, AllFeaturesMean(y,f) \neq AllFeaturesMean\,(\acute{y},f)\ for\ all\ \acute{y} \in Y, \acute{y} \neq y$$

Where:

- *SelectedFeatures(y)* is the set of features within class $y$ after filtering out those with equal mean values.

- *AllFeaturesMean(y,f)* represents the mean value of feature $f$ within class $y$.

- $\acute{y} \neq y$ ensures that we are comparing distinct classes.

- The condition $AllFeaturesMean\,(y,f) \neq AllFeaturesMean\,(\acute{y},f)$ checks if the mean values of features f are not equal across different classes.

  To represent the Selected features for all classes:

$$SelectedFeaturesAllClasses = \bigcap_{y \in Y} SelectedFeatures\,(y)$$

Where:

- *SelectedFeaturesAllClasses* is the set of features that have distinct mean values across all classes.

- $\cap$. represent the intersection operation over all classes.

**2.5.5. Extreme Boosting (XGBoost) model for classification.** The combined static and dynamic feature set was input into an XGBoost model for multiclass classification. XGBoost [57], an ensemble learning algorithm, constructs multiple models sequentially, with each subsequent model addressing the shortcomings of its predecessors. This approach, rooted in gradient boosting, incorporates regularization to prevent overfitting and accommodate various loss functions [58].

XGBoost traditionally uses convex loss functions, recent research has explored custom and non-convex loss functions to enhance performance in specific applications [59]. For instance, [60] investigated the use of squared logistics loss (SqLL) to improve accuracy. [59] developed weighted softmax loss functions for industrial applications, while [61] proposed a generalized XGBoost method accommodating both convex and some non-convex loss functions. These advancements demonstrate XGBoost's versatility and potential for tailored solutions in various

domains, including big data analysis and multi-objective parameter regularization. The purpose of the XGBoost classifier is multifaceted and versatile, as evidenced by various research studies. XGBoost is utilized for enhancing prediction accuracy in diverse fields such as meteorology for hailstorm forecasting [62], detecting patterns in financial datasets to differentiate between solvable and bankrupt situations [63], improving learner performance prediction in Intelligent Tutoring Systems by enhancing models like Performance Factor Analysis and DAS3H [64], and detecting malware in Internet of Medical Things (IoMT) data for better medical assistance through dimensionality reduction and efficient classification [65]. The XGBoost algorithm's scalability, robustness, and proficiency with complex datasets make it a valuable tool for increasing prediction accuracy, addressing class imbalances, enhancing performance prediction models, and improving data analysis in various domains.

By building upon these principles, XGBoost has demonstrated superior performance in tasks such as lung cancer detection. While traditional gradient boosting involves a single optimization step, XGBoost employs a two-stage approach. This separation aims to improve both the optimization process itself and the selection of the step direction. But the XGBoost solve,

$$\frac{\partial S(y, f^{(m-1)}(x) + f_m(x))}{\partial f_m(x)} = 0 \tag{1}$$

For every x in data to directly fix the step. We have,

$$S\left(y, f^{(m-1)}(x) + f_m(x)\right) \tag{2}$$

$$\approx S\left(y, f^{(m-1)}(x)\right) + g_m(x)f_m(x) + \frac{1}{2}h_m(x)f_m(x)^2 \tag{3}$$

$$\approx S\left(y, f^{(m-1)}(x)\right) + g_m(x)f_m(x) + \frac{1}{2}h_m(x)f_m(x)^2 \tag{4}$$

Leveraging the second-order Taylor expansion to approximate the loss function, where $g_m$ $(x)$ is gradient and $h_m(x)$ is Hessian.

$$h_m(x) = \frac{\partial^2 S(Y, f(x))}{\partial f(x)^2}, \quad here \ f(x) = f^{(m-1)}(x)$$

Then, loss function can be rewritten as:

$$S(f_m) \approx \sum_{i=1}^{n}\left[g_m(x_i)f_m(x_i) + \frac{1}{2}h_m(x_i)f_m x^2\right] + \ const \tag{5}$$

$$\propto \ \sum_{j=1}^{P_m}\sum_{i \in R_{jm}}\left[g_m(x_i)K_{jm} + \frac{1}{2}h_m(x_i)K_{jm}^2\right] \tag{6}$$

In region j, lets $G_{jm}$ denotes sum of gradient and the sum of Hessian is represented by $H_{jm}$, then equation will be,

$$S(f_m) \propto \sum_{j=1}^{P_m}\left[G_{jm}K_{jm} + \frac{1}{2}H_{jm}K_{jm}^2\right] \tag{7}$$

The optimal value can be computed using below function

$$K_{jm} = -\frac{G_{jm}}{H_{jm}}, \text{ where } j = 1, 2, \ldots, P_m \tag{8}$$

We get loss function when we plug it back

$$S(f_m) \propto -\frac{1}{2} \sum_{j=1}^{P_m} \frac{G^2_{jm}}{H_{jm}} \tag{9}$$

The tree structure is marked using this function. The lesser the score indicates better structure [57]. The maximum gain for every split is:

$$gain = \frac{1}{2} \left[ \frac{G^2_{jm\ Left}}{H_{jm\ Left}} + \frac{G^2_{jm\ Right}}{H_{jm\ Right}} - \frac{G^2_{jm}}{H_{jm}} \right] \tag{10}$$

Which is,

$$gain = \frac{1}{2} \left[ \frac{G^2_{jm\ Left}}{H_{jm\ Left}} + \frac{G^2_{jm\ Right}}{H_{jm\ Right}} - \frac{\left( G_{jm\ Left} + G_{jm\ Right} \right)^2}{H_{jm\ Left} + H_{jm\ Right}} \right]. \tag{11}$$

For improved performance, we can rewrite the loss function as follows, incorporating regularization criteria:

$$S(f_m) \propto \sum_{j=1}^{P_m} \left[ G_{jm} K_{jm} + \frac{1}{2} H_{jm} K^2_{jm} \right] + \gamma P_m + \frac{1}{2} \lambda \sum_{j=1}^{P_m} K^2_{jm} + \alpha \sum_{j=1}^{P_m} \left| K_{jm} \right| \tag{12}$$

$$= \sum_{j=1}^{P_m} \left[ G_{jm} K_{jm} + \frac{1}{2} \left( H_{jm} + \lambda \right) K^2_{jm} + \alpha \left| K_{jm} \right| \right] + \gamma P_m \tag{13}$$

Where $\gamma$ penalizes the number of leave, $\alpha$ denotes L1 regularization while $\lambda$ denotes L2 regularization. The optimal weight can calculate for each region j as:

$$K_{jm} = \begin{cases} -\dfrac{G_{jm} + \alpha}{H_{jm} + \lambda} & G_{jm} < -\alpha \\[2ex] -\dfrac{G_{jm} - \alpha}{H_{jm} + \lambda} & G_{jm} > \alpha \\[2ex] 0 & else \end{cases} \tag{14}$$

And the gain is,

$$gain = \frac{1}{2} \left[ \frac{P_\alpha \left( G^2_{jm\ Left} \right)}{H_{jm\ Left} + \lambda} + \frac{P_\alpha \left( G^2_{jm\ Right} \right)}{H_{jm\ Right} + \lambda} - \frac{P_\alpha \left( G_{jm} \right)^2}{H_{jm} + \lambda} \right] - \gamma \tag{15}$$

Where,

$$P_\alpha(G) = \begin{cases} G + \alpha & G < -\alpha \\ G - \alpha & G > \alpha \\ 0 & else \end{cases} \tag{16}$$

The XGBoost classifier stands out for several reasons. It offers a rich set of randomization and regularization options during the learning process, which helps to prevent overfitting and improve model generalizability. Additionally, XGBoost boasts faster training times and user-friendliness. To leverage these advantages in our study, we employed the following hyperparameters as reflected in Table 1.

The core challenge in optimizing the loss function is finding the minimum value, which can be local or global depending on the function's shape (e.g., quadratic functions). To address overfitting, XGBoost introduces new regularization features, enhancing its ability to resist this common problem. The detailed structure of XGBoost is illustrated in Fig 4.

## 2.6. Performance evaluations measures

We employ standard performance evaluation metrics as outlined in [66]:

### 2.6.1. Precision.

$$Precion = \frac{\text{Number of relevant items retreived}}{\text{Number of retrieved items}} = P(\text{relevent} \mid \text{retreived})$$

### 2.6.2. Recall.
Recall (R) represents the proportion of relevant documents that the model successfully retrieves out of all the relevant documents in the dataset.

$$Recall = \frac{\text{Number of relevant items retreived}}{\text{Total Nuber of Relevent Document}}$$

### 2.6.3. F-measure.
The F1-measure calculation treats each record as a query-class pair. In this context, each class represents the desired documents for the query (record), and we compute both recall and precision for each class within that record. The $F_1$-measure of record $j$

**Table 1. Hyperparameters optimization of XGBoost algorithm.**

| Model | Hyperparameters | Tuned Parameters |
|---|---|---|
| **XGBoost** | 1- booster<br> -: gbtree, gblinear<br>2- colsample_bytree<br> -: 0.4, 0.6, 0.8, 1<br>3- learning_rate<br> -: 0.01, 0.1, 0.2, 0.4<br>4- max_depth<br> -: 2, 3, 4, 6<br>5- n_estimators<br> -: 200, 300, 400, 500<br>6- subsample<br> -: 0.4, 0.6, 0.8, 1 | 1- subsample<br> -: 0.8<br>2- n_estimators<br> -: 200<br>3- max_depth<br> -: 6<br>4- learning_rate<br> -: 0.1<br>5- colsample_bytree<br> -: 1<br>6- booster<br> -: gbtree |

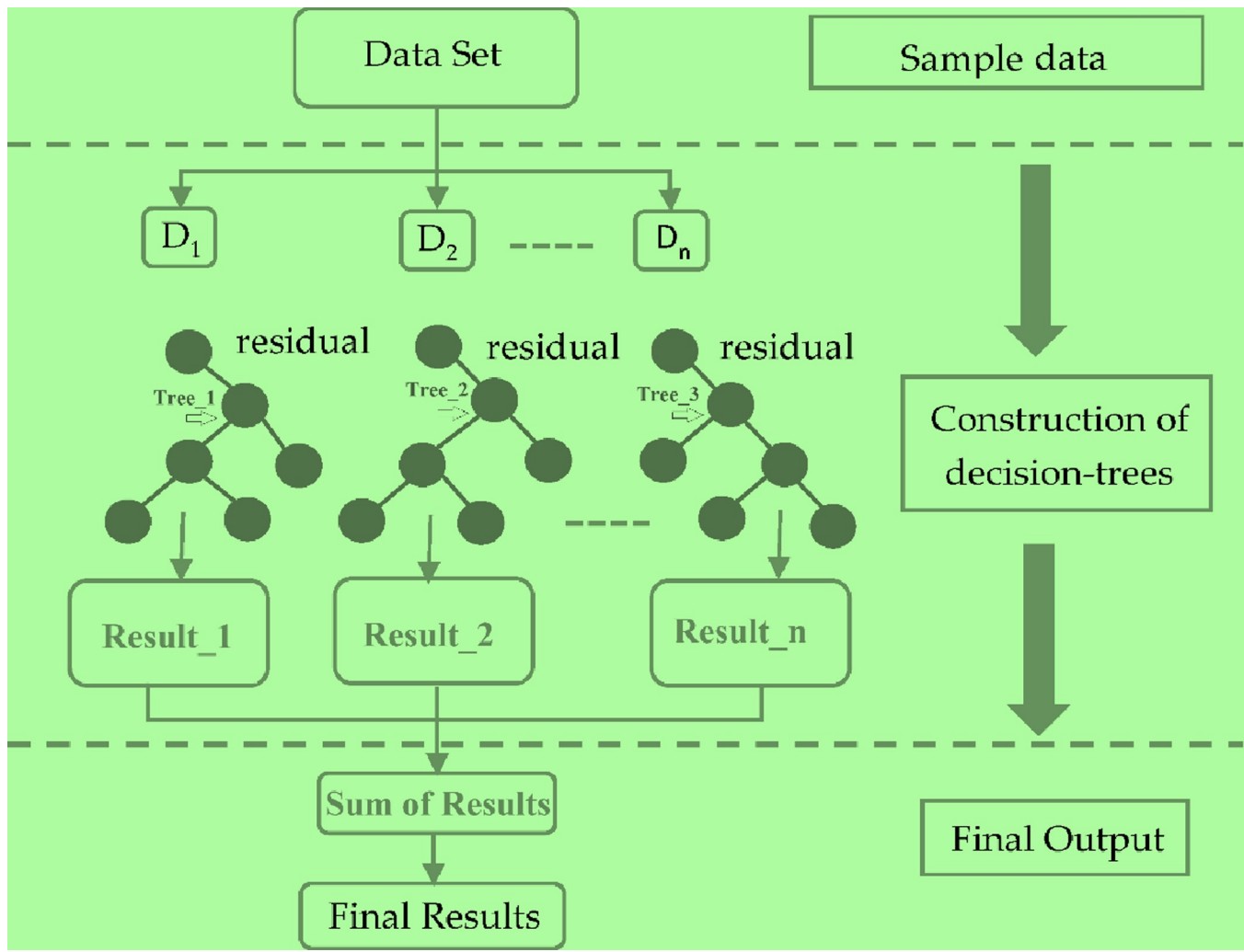

**Fig 4. General architecture of XGBoost algorithm.**

and class *i* is defined as follows:

$$F_{ij} = \frac{2*\text{Recall}(i,j)*\text{precion}(i,j)}{\text{Recall}(i,j) + \text{precion}(i,j)}$$

**2.7. Receiver operating characteristic (ROC) curve.** To assess our classifier's ability to distinguish between COVID-19 and non-COVID-19 cases, we employed sensitivity (True Positive Rate) and specificity (1-False Positive Rate). We assigned binary labels to cases and generated a Receiver Operating Characteristic (ROC) curve. This curve plots sensitivity against specificity. The ROC curve's shape and the Area Under the Curve (AUC) quantify the classifier's performance. Higher AUC indicates better separation between the two classes. Sensitivity reflects the proportion of correctly identified COVID-19 cases, while specificity reflects the proportion of correctly identified non-COVID-19 cases [67]. The ROC curve and AUC provide a visual and numerical assessment of the classifier's ability to differentiate between the disease and healthy cases [68].

## 3. Results

This section presents a detailed analysis of the proposed ESN-MDFS model's performance through confusion matrices, tabular data, and AUC values. Additionally, a comparative evaluation with existing studies is provided.

Fig 5 presents the multi-class COVID-19 detection results exclusively based on VGG-16 deep features, as visualized through confusion matrices (5a), classification reports (5b), AUC curves (5c), and accuracy loss curves (5d). Relying solely on deep features, the model achieved

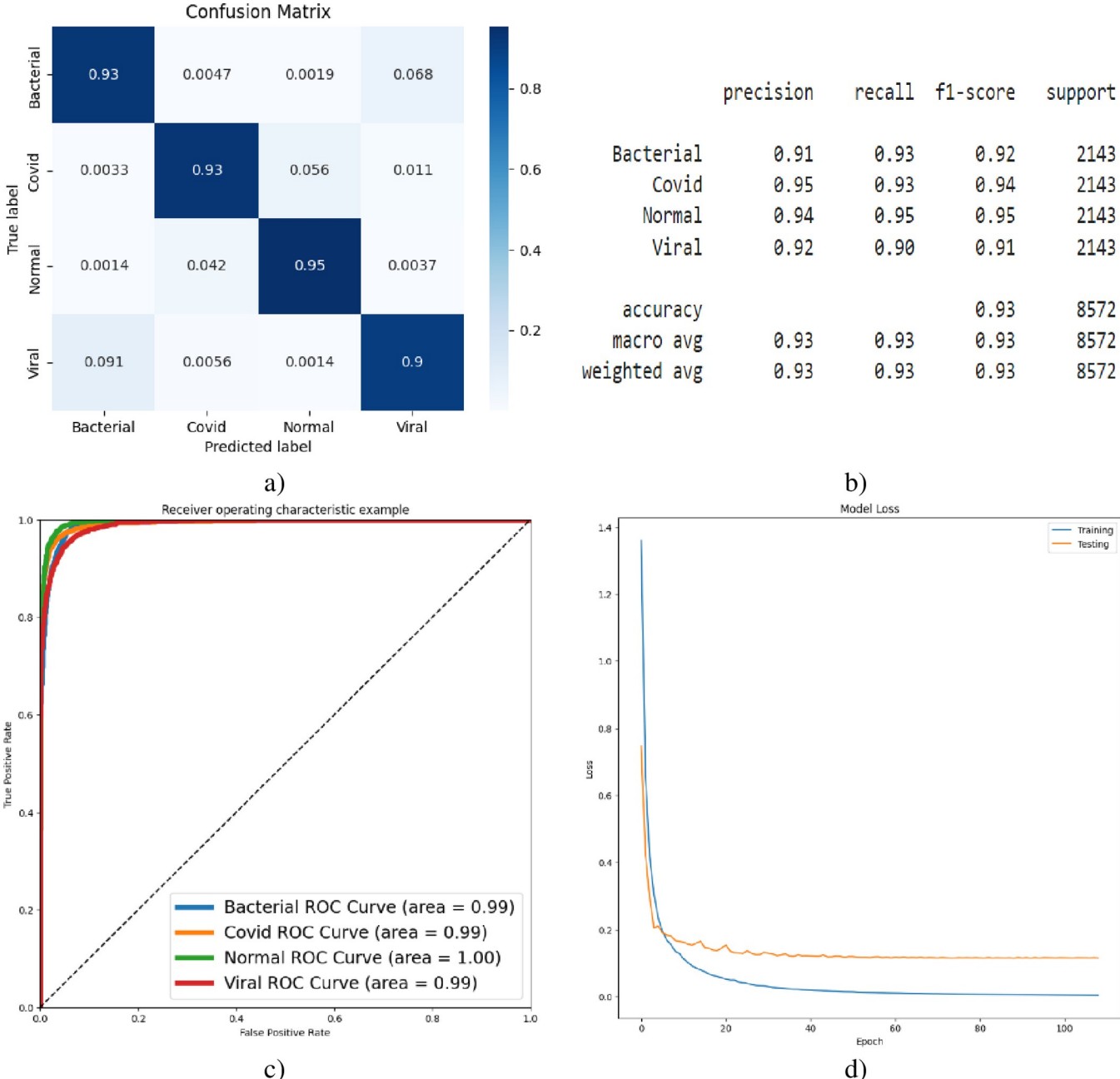

**Fig 5.** Multi-class Covid-19 detection using VGG-16, a) Confusion Matrix, b) Classification Report, c) AUC, d) Accuracy-Loss Curve.

an overall accuracy of 93% in classifying the four target classes. Notably, the AUC for multi-class differentiation (bacterial, COVID-19, viral) was 0.99, while perfect discrimination (AUC of 1.00) was observed for the normal class.

Fig 6 illustrates the multi-class COVID-19 detection performance solely based on XGBoost-processed static features, as depicted in the confusion matrix (6a), classification report (6b), and AUC curve (6c). Relying exclusively on static features, the model achieved an overall accuracy of 86% in classifying the four target classes.

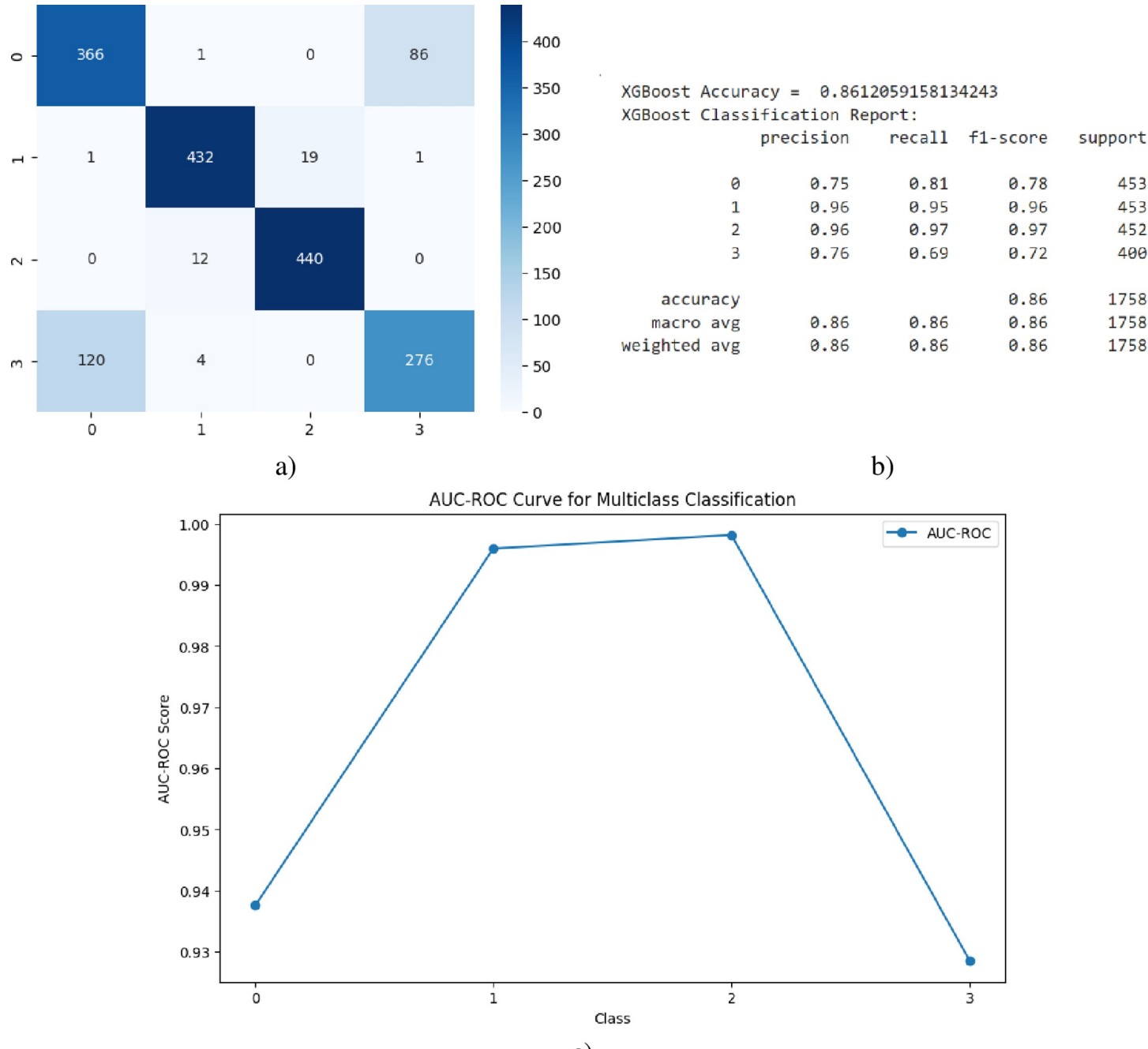

**Fig 6.** Multi-class Covid-19 detection using XGBoost, a) Confusion Matrix, b) Classification Report, c) AUC, d) Accuracy-Loss Curve.

The **Fig 7** reflected the confusion matrix distinguish the four classes by utilizing Multiclass (COVID-19, normal, bacterial, viral pneumonia,) detection by utilizing Intelligent Extreme Smart Deep Network without mean dropout.

The Table 2 reflects the multiclass classification performance by utilizing Intelligent Extreme Smart Deep Network Hybrid Feature Space without mean dropout. The model achieved an impressive accuracy of 95.54% in classifying images into four classes: Bacterial, COVID-19, Normal, and Viral. This indicates that the model correctly classified 95.54% of the images in the dataset. To detect the Bacteria, the model demonstrated excellent performance with a precision, recall, and F1-score of 99.20%, 98.21%, and 98.70% respectively. This suggests that the model is highly effective in correctly identifying bacterial infections. To predict COVID-19, while the performance for COVID-19 is still good, it is slightly lower compared to the other classes. The precision, recall, and F1-score are 93.66%, 90.69%, and 92.15% respectively. This indicates that there might be some room for improvement in accurately identifying COVID-19 cases. To detect the normal subject, the model achieved a precision of 92.10%, recall of 94.84%, and F1-score of 93.45% for the normal class. These metrics suggest reasonable

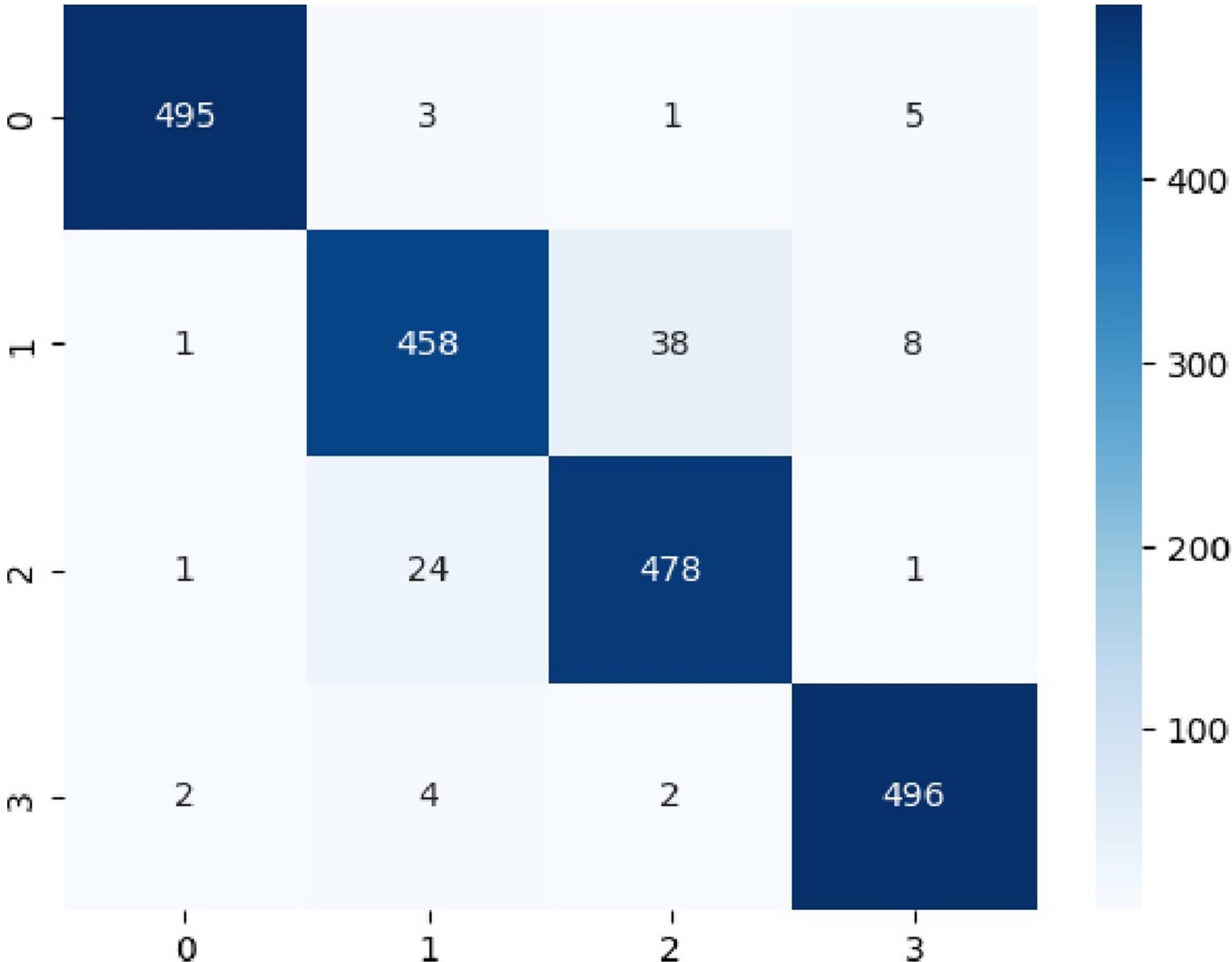

**Fig 7. Confusion Matrix to distinguish Multiclass COVID-19 detection by utilizing ESN: Extreme Smart Network without MeanDropout feature space.**

**Table 2. Multiclass COVID-19 detection by utilizing ESN: Extreme Smart Network without MeanDropout feature space.**

| Class | Precision | Recall | F1-Score | Support |
|---|---|---|---|---|
| Bacterial | 99.20% | 98.21% | 98.70% | 504 |
| COVID-19 | 93.66% | 90.69% | 92.15% | 505 |
| Normal | 92.10% | 94.84% | 93.45% | 504 |
| Viral | 97.25% | 98.41% | 97.83% | 504 |
| Accuracy | | | 95.54% | 2017 |
| Macro Avg | 95.55% | 95.54% | 95.53% | 2017 |
| Weighted Avg | 95.55% | 95.54% | 95.53% | 2017 |

performance in identifying normal cases. To predict the viral, similar to bacterial, the model showed excellent performance in classifying viral infections with a precision, recall, and F1-score of 97.25%, 98.41%, and 97.83% respectively.

The **Fig 8** reflected the confusion matrix distinguish the four classes by utilizing the proposed ESN-MDFS Covid model.

The Table 3 reflects the multiclass classification performance by using ESN-MDFS Covid model. The model achieved an impressive accuracy of 96.18% in classifying images into four classes: Bacterial, COVID-19,Normal, and Viral. This indicates that the model correctly classified 96.18% of the images in the dataset. For Bacterial and Viral, the model demonstrated exceptional performance with precision, recall, and F1-score values close to 98% for both classes. This suggests that the model is highly effective in correctly identifying bacterial and viral infections. To detect COVID-19 and Normal, while the performance for COVID-19 and Normal classes is also good, it is slightly lower compared to Bacterial and Viral. The model achieved precision, recall, and F1-score values around 94% for both classes, indicating room for improvement in accurately differentiating between these two classes. The results suggest that the ESN-MDFS model is a promising approach for classifying lung conditions from medical images. It achieves high accuracy and performs well across all four classes, even better than the model presented in Table 2.

The Fig 9 reflects the accuracy-loss graph for multi-class Covid-19 detection at 150 epochs and using the proposed ESN-MDFS Covid model. The highest AUC of 0.99 was yielded to detect bacterial and viral pneumonia followed by AUC of 0.96 to detect normal and AUC of 0.95 to detect the COVID-19 from multiclass.

Fig 10 illustrates the accuracy of the ESN-MDFS model across seven cross-validation folds. The model achieved a mean accuracy of 95.57% with a standard deviation of 0.54, demonstrating consistent performance across different data subsets.

Table 4 presents a comparison of the proposed ESN-MDFS model with several existing lightweight models for multiclass COVID-19 detection. The comparison focuses on model size and accuracy. For Model Size, there's a significant difference in model sizes, with the proposed ESN-MDFS being significantly smaller (889 KB) compared to other models. To compute accuracy, the proposed ESN-MDFS model outperforms all other lightweight models in terms of accuracy (96.18%). Trade-off between Size and Accuracy: While larger models like DL trained Model Size and EfficientNetV2L offer higher accuracy, they also demand significantly more computational resources. ESN-MDFS model demonstrates a compelling balance between model size and accuracy, making it a potential candidate for deployment on resource-constrained devices.

This study identified redundant information within the static features, necessitating parameter reduction. The high dimensionality of GLCM-extracted texture features, especially when considering multiple distances and angles, posed a significant challenge. To address this, we introduced ESN-MDFS, which employs Mean Dropout to refine the hybrid feature space

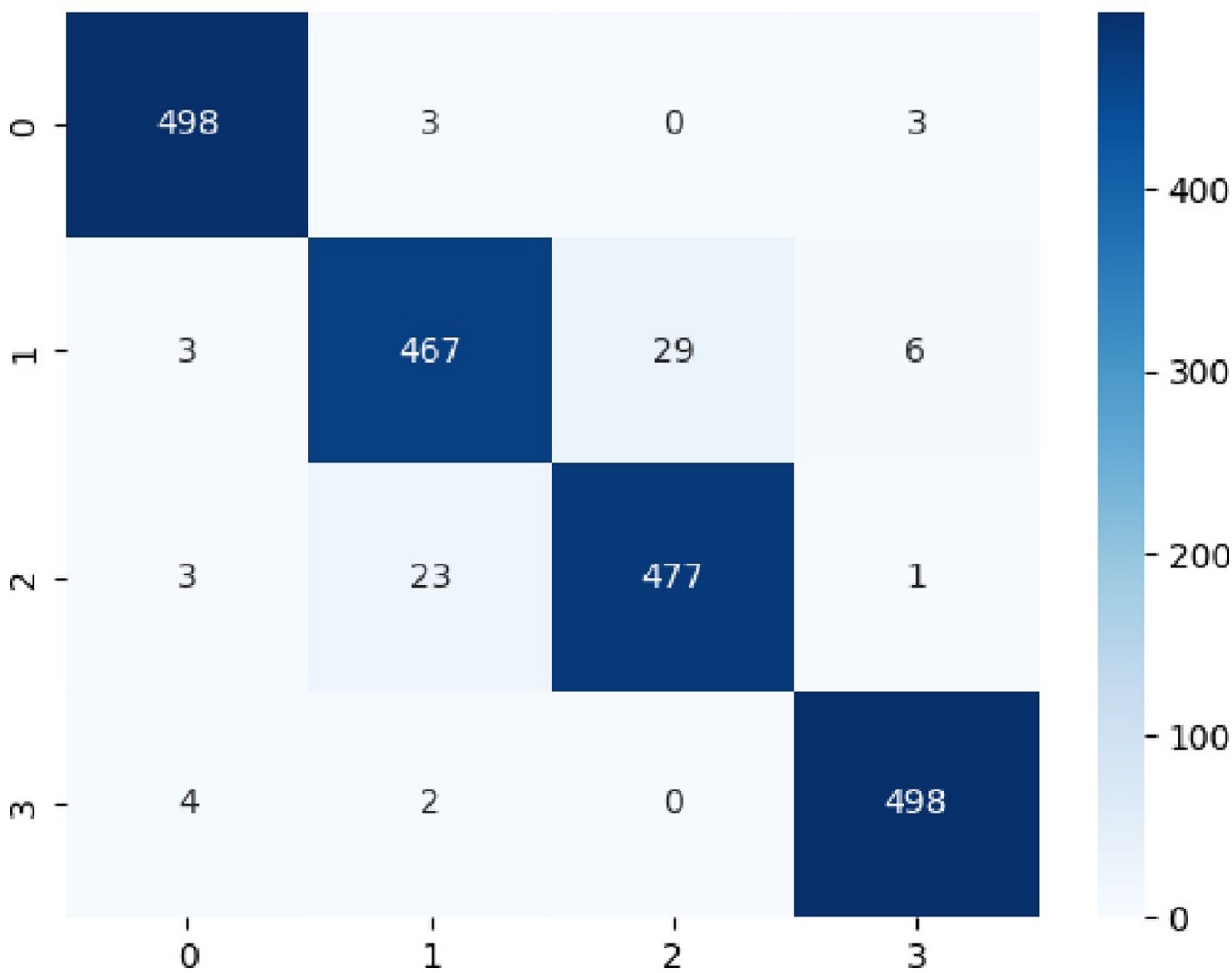

**Fig 8. Confusion matrix to distinguish multiclass COVID-19 detection by utilizing ESN-MDFS: Extreme Smart Network using mean dropout feature selection technique.**

(HFS) by eliminating less informative features. This approach yielded a substantial reduction in model size, resulting in a remarkable accuracy of 96.18%. The resulting lightweight model prioritizes performance and efficiency, making it ideal for resource-constrained edge devices. Reduced storage requirements, faster computation, and lower power consumption are key advantages of this compact architecture.

## 4. Discussions

The COVID-19 pandemic has led to a global health crisis, marked by millions of confirmed cases and substantial mortality rates. While numerous studies have explored the application of Convolutional Neural Networks (CNNs) for COVID-19 classification using chest X-rays and CT scans, most research has been limited to binary comparisons, differentiating COVID-19 from pneumonia or normal conditions. This binary approach falls short of the diagnostic complexities often inherent in infectious diseases.

**Table 3. Multiclass COVID-19 detection by utilizing ESN-MDFS: Extreme Smart Network using mean dropout feature selection technique.**

| Class | Precision | Recall | F1-Score | Support |
|---|---|---|---|---|
| Bacterial | 98.03 | 98.81% | 98.42% | 504 |
| COVID-19 | 94.34% | 92.48% | 93.40% | 505 |
| Normal | 94.27% | 94.64% | 94.46% | 504 |
| Viral | 98.03% | 98.81% | 98.42 | 504 |
| Accuracy | | | 96.18% | 2017 |
| Macro Avg | 96.17% | 96.18% | 96.17% | 2017 |
| Weighted Avg | 96.17% | 96.17% | 96.17% | 2017 |

Table 5 presents a comparative analysis of several studies focused on COVID-19 lung infection detection using AI-based methods. The comparison encompasses key factors such as modality (X-ray or CT), dataset size, methodology, and performance metrics. Regarding modality, most studies utilized X-ray imaging for analysis, except for Ying et al., which employed CT scans. For Dataset Size, there's a significant variation in dataset sizes across studies, ranging from relatively small datasets (Sethy et al.) to larger ones (Ying et al. and this study). Regarding methodology, a diverse range of methods was employed, including CNN, ResNet50, SVM, DRE-Net, texture features with machine learning, and the proposed ESN-MDFS. To compute performance, the proposed ESN-MDFS method achieved the highest accuracy (96.18%) among the compared studies, surpassing other methods in terms of overall classification performance. Ghoshal et al. and Sethy et al. used smaller datasets and simpler models, resulting in lower accuracy compared to the proposed method. Ying et al. used CT scans, which provide more detailed information than X-rays, but still achieved a lower accuracy than the proposed method. Hussain et al. focused on two-class classification, while this study addressed a more complex four-class classification problem and achieved higher accuracy.

The primary outcome measured in this study [73] is the accuracy of COVID-19 detection using CT-scan images and various preprocessing methods. The main findings of this study are that different preprocessing methods, including resizing, enhancement, and normalization, had an impact on the accuracy of COVID-19 classification using a deep learning model (VGG-16), and the highest accuracy of 88.54% was achieved using a combination of deformed resizing, CLAHE enhancement, and normalization to the range of [0 1] and [-1 1].

The proposed ESN-MDFS model surpasses existing methods in accurately classifying multiclass COVID-19 infections. By integrating Mean Dropout Feature Selection, the model effectively balances performance and computational efficiency. Leveraging X-ray imaging, the model effectively differentiates between COVID-19, bacterial pneumonia, viral pneumonia, and normal conditions. Demonstrating exceptional accuracy across various lung infection types, the model's compact size makes it suitable for resource-constrained environments. Although not explicitly evaluated, the model's strong performance suggests potential adaptability to diverse datasets and clinical contexts.

## 5. Conclusion

Our proposed ESN-MDFS model significantly advances COVID-19 detection by accurately differentiating chest X-rays into four categories: COVID-19, bacterial pneumonia, viral pneumonia, and normal. This multi-class classification system has the potential to revolutionize patient care by streamlining clinical workflows, enabling early diagnosis, optimizing patient triage, and facilitating disease progression monitoring. These capabilities position ESN-MDFS

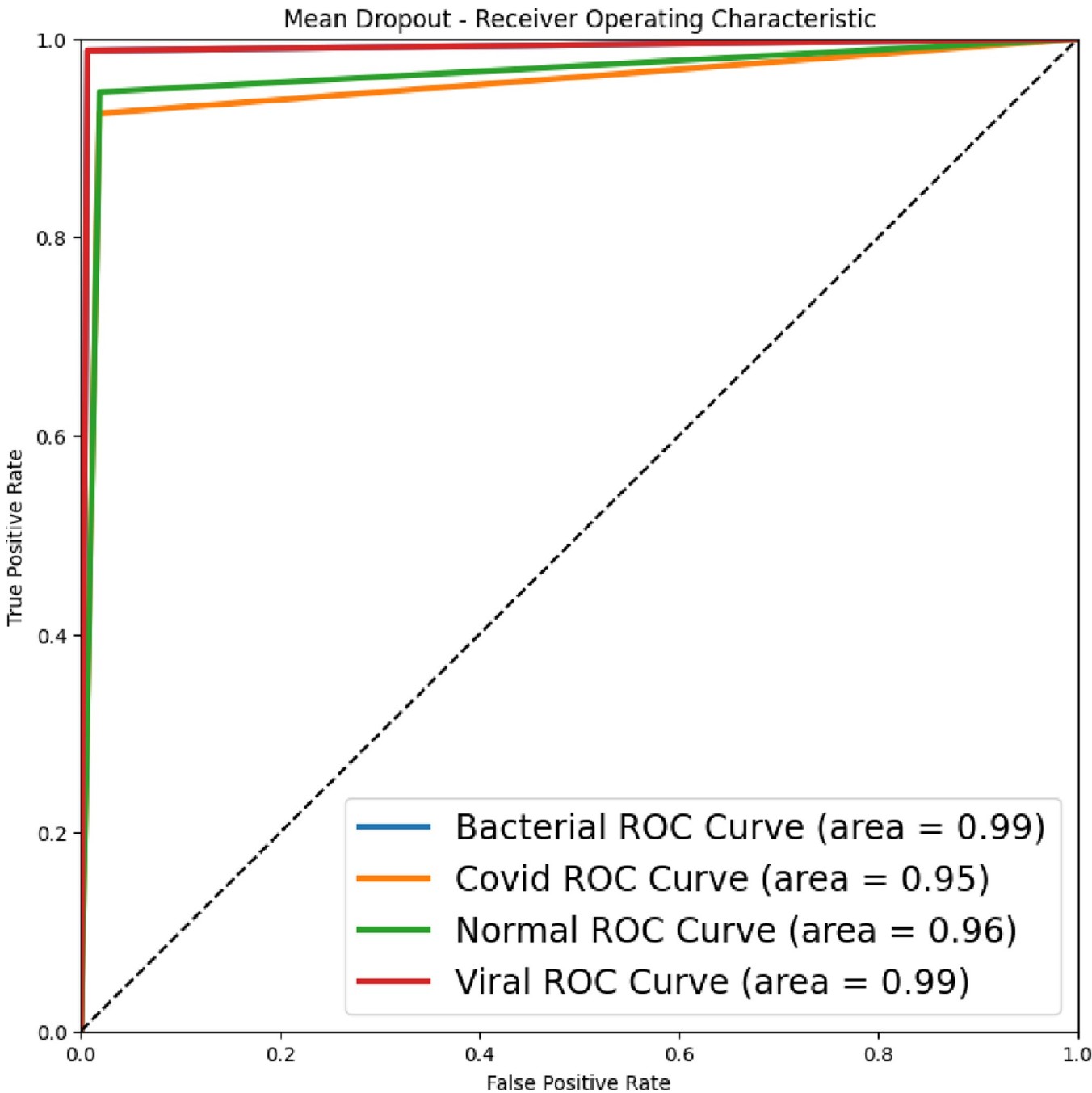

**Fig 9. Area under the receiver operating characteristic curve (AUC) to distinguish Multiclass COVID-19 detection by utilizing ESN-MDFS.**

as a critical tool in combating COVID-19. This innovative approach maintains high model accuracy while drastically reducing memory footprint, making it suitable for resource-constrained edge devices. Deploying this optimized model enables real-time, point-of-care lung nodule detection, eliminating the need for centralized servers. Clinicians can benefit from immediate diagnostic insights, facilitating faster treatment decisions and improved patient outcomes. Beyond accuracy, this approach streamlines workflows by automating chest X-ray analysis, reducing diagnostic turnaround times, and enhancing overall efficiency. Early disease

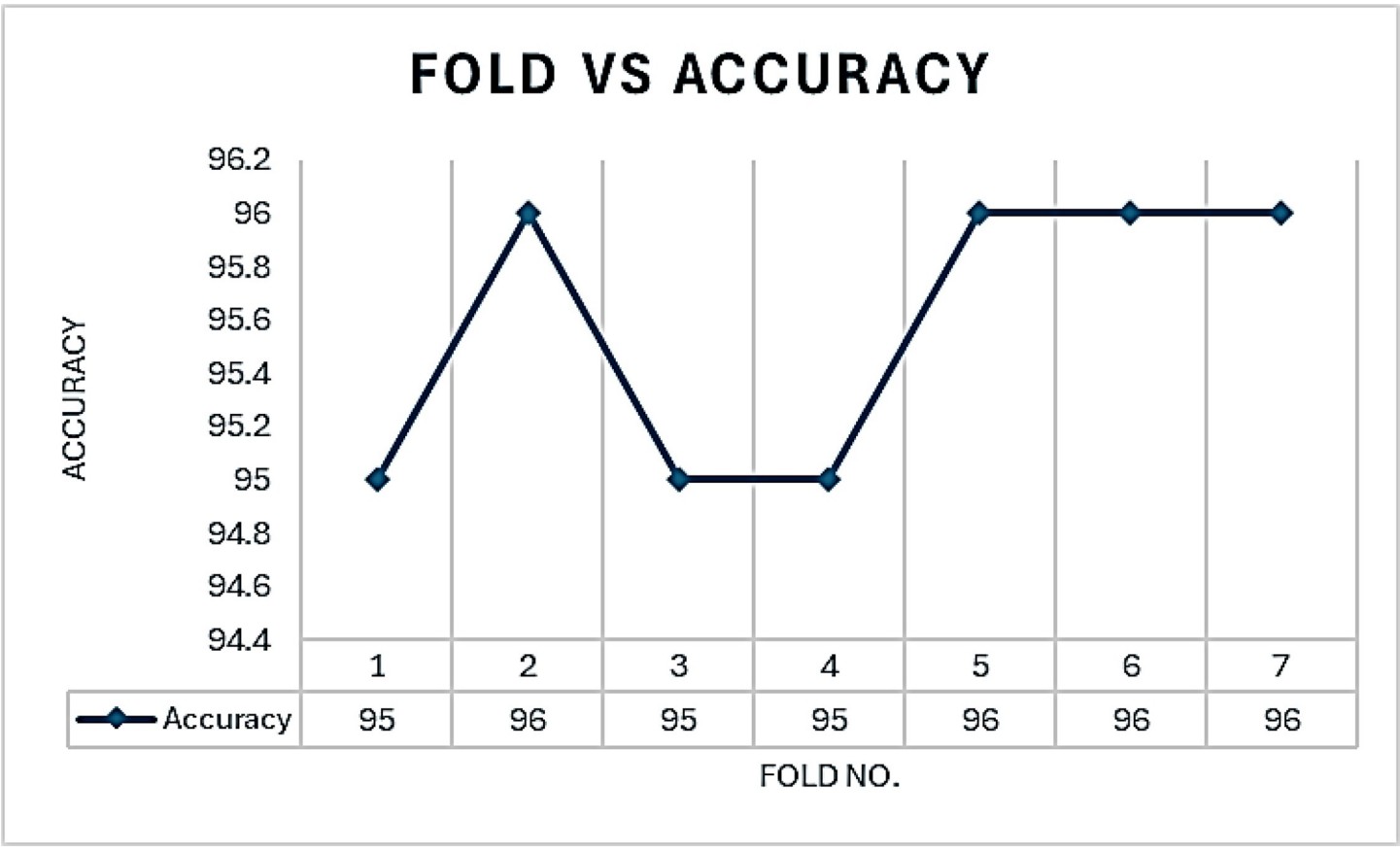

**Fig 10. Fold vs accuracy curve to distinguish multi-class using ESN-MDFS.**

detection, particularly for conditions like COVID-19, is facilitated by the model's improved sensitivity and accuracy. Moreover, the ability to differentiate between various pneumonia types enables effective patient triage and resource allocation. By tracking changes in chest X-ray features over time, clinicians can gain valuable insights into disease progression and tailor treatment strategies accordingly.

## 5.1. Limitations and future directions

While the ESN-MDFS model demonstrates promising results, several limitations and opportunities for improvement exist. The model's performance is influenced by dataset quality,

**Table 4. Comparison of Multiclass COVID-19 detection by utilizing ESN-MDFS with existing Lightweight models.**

| S# | Size comparison (Non-Lightweight VS Lightweight) | | Accuracy comparison (Non-Lightweight VS Lightweight) | |
|---|---|---|---|---|
| | Model | Size | Model | Accuracy |
| 1 | DL trained Model Size | 616 MB | DL trained Model Size | 95.54% |
| 2 | EfficientNetV2L | 455 MB | EfficientNetV2L | 73.00% |
| 3 | ConvNeXtTiny | 108 MB | ConvNeXtTiny | 72.00% |
| 4 | MobileNetV2 | 12 MB | MobileNetV2 | 67.00% |
| 5 | Proposed ESN-MDFS | 889 KB | Proposed ESN-MDFS | 96.18% |

**Table 5. Comparison of AI-assisted recent studies for COVID-19 lung infection.**

| Authors | Modality | Subjects | Method | Performance |
|---|---|---|---|---|
| Ghoshal et al. [69] | X-Ray | COVID-19 90 and other conditions | CNN | 92.9% (Acc.) |
| Sethy et al. [70] | X-ray | COVID-19 and Normal 25 images | ResNet50 and SVM | 95.33%(Acc.) |
| Ying et al. [71] | CT | COVID-19 777 images and 708 images of Normal | DRE-Net | 86% (Acc.) |
| Hussain et al. [72] | X-Ray | COVID-19 Bacterial & Viral 145 images and 138 Normal | Texture features using Machine learning. Two-class classification i) covid-19 vs normal ii) Covid-19 vs viral pneumonia iii) Covid-19 vs Bacterial pneumonia iv) Four-class (Covid-19, Bacteria, Viral and Normal) | 100% accuracy 97.56% Accuracy 97.44% Accuracy 79.52% Accuracy |
| Pratiwi et al. [73] | CT | Covid—(1251) Non-Covid–(1229) | Two Classes Deep learning VGG-16 | 88.54% Accuracy |
| This study | X-Ray | COVID-19 (N = 1525), non-COVID-19 normal (N = 1525), viral pneumonia (N = 1342) and bacterial Pneumonia (N = 2521) After augmentation N = 2521 | 4-class (Normal, Bacterial Pneumonia, viral Pneumonia and COVID-19) using ESN-MDFS approach | 96.18% Accuracy AUC of 0.99 |

diversity, and image acquisition protocols. Although GLCM features enhance performance, manual feature engineering is time-consuming. Additionally, the model's black-box nature hinders interpretability and clinical adoption. To address these challenges, future research should focus on expanding the dataset, automating feature extraction, improving model interpretability, incorporating additional data modalities, optimizing for real-time performance, and conducting rigorous benchmarking. By pursuing these directions, the model's potential can be fully realized.

## Acknowledgments

Ashit Kumar Dutta would like to express sincere gratitude to AlMaarefa University, Riyadh, Saudi Arabia, for providing funding to conduct this research.

## Author Contributions

**Conceptualization:** Saghir Ahmed, Basit Raza, Lal Hussain.

**Data curation:** Saghir Ahmed, Basit Raza, Lal Hussain.

**Formal analysis:** Saghir Ahmed, Basit Raza, Lal Hussain, Touseef Sadiq, Ashit Kumar Dutta.

**Investigation:** Saghir Ahmed, Basit Raza, Lal Hussain, Touseef Sadiq, Ashit Kumar Dutta.

**Methodology:** Saghir Ahmed, Basit Raza, Lal Hussain, Touseef Sadiq.

**Project administration:** Saghir Ahmed, Lal Hussain.

**Resources:** Saghir Ahmed, Lal Hussain.

**Software:** Saghir Ahmed, Basit Raza, Lal Hussain, Ashit Kumar Dutta.

**Supervision:** Basit Raza, Lal Hussain.

**Validation:** Saghir Ahmed, Basit Raza, Lal Hussain, Touseef Sadiq.

**Visualization:** Saghir Ahmed, Basit Raza, Lal Hussain, Touseef Sadiq.

**Writing – original draft:** Saghir Ahmed, Ashit Kumar Dutta.

**Writing – review & editing:** Lal Hussain, Touseef Sadiq, Ashit Kumar Dutta.

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
