## [Decision Letter · Decision Letter 0]

22 Jul 2024

PONE-D-24-21569Enhancing Multiclass COVID-19 Prediction with ESN-MDFS: Extreme Smart Network using Mean Dropout Feature Selection TechniquePLOS ONE

Dear Dr. Hussain,

Thank you for submitting your manuscript to PLOS ONE. After careful consideration, we feel that it has merit but does not fully meet PLOS ONE’s publication criteria as it currently stands. Therefore, we invite you to submit a revised version of the manuscript that addresses the points raised during the review process.

We look forward to receiving your revised manuscript.

Kind regards,

Catalin Buiu

Academic Editor

PLOS ONE

Journal Requirements:

Reviewers' comments:

Reviewer's Responses to Questions

**Comments to the Author**

1. Is the manuscript technically sound, and do the data support the conclusions?

Reviewer #1: Yes

Reviewer #2: Yes

Reviewer #3: Yes

2. Has the statistical analysis been performed appropriately and rigorously? 

Reviewer #1: I Don't Know

Reviewer #2: Yes

Reviewer #3: Yes

3. Have the authors made all data underlying the findings in their manuscript fully available?

Reviewer #1: No

Reviewer #2: Yes

Reviewer #3: No

4. Is the manuscript presented in an intelligible fashion and written in standard English?

Reviewer #1: Yes

Reviewer #2: No

Reviewer #3: Yes

5. Review Comments to the Author

Reviewer #1: Authors should address the following major revisions:

1) These research, by virtue of the fact, stress the existence and severity of dire results associated with delayed and improper diagnosis which can be a real threat to the survival of

humankind. The latest studies have revealed that deep learning models can be used for coronavirus image classification.- add the following work for this statement such as: A Deep learning-based X-Ray Imaging Diagnosis System for Classification of Tuberculosis, COVID19, and Pneumonia Traits using Evolutionary Algorithm

2) While deep learning models have shown promise in classifying COVID-19 images, there are still limitations that need to be addressed. One such limitation is the lack of research on applying domain adaptation techniques to overcome the challenge of the cross-dataset problem.- add the following work for this statement such as: Deep learning network selection and optimized information fusion for enhanced COVID-19 detection

3) Related work of this manuscript can be further enhanced by adding few recent works. Also, discuss the cutting edges and gaps that should be linked with the proposed work.

4) How many features are extracted from the VGG model? How many layers used for the re-training on the selected datasets?

5) how many parameters are trained for the VGG model and how the hyperparameters are selected? add in the manuscript.

6) What is the purpose of GLCM features and how many features are extracted?

7) What is the loss function of the XGBosst classifier and what is the purpose of this classifier?

Reviewer #2: The paper proposes a novel approach called “Enhancing COVID Prediction with ESN-MDFS”, which combines an Extreme Smart Network (ESN) and a Mean Dropout Feature Selection Technique (MDFS) to improve the multi-class detection of COVID-19 and three other lung conditions (normal, viral, and bacterial) using portable chest x-rays (CXRs). This method extracts both static features using Grey-level Co-occurrence Matrix (GLCM) texture analysis and dynamic features through a VGG-16 pre-trained deep learning model.

The authors highlight the limitations of traditional diagnostic methods, such as RT-PCR and biomarkers. Their study seeks to overcome challenges like data variability and the need for large labeled datasets by developing a robust, lightweight deep learning model for COVID-19 detection.

The authors present the key steps of the proposed method, namely:

1. Preprocessing: Images are resized, denoised using median blur, and normalized for intensity.

2. Data Augmentation: Techniques like rotation, flipping, shearing, contrast adjustments, and various transforms are applied to balance the dataset and prevent overfitting.

3. Feature Selection: Static Features are extracted using Grey Level Co-occurrence Matrix (GLCM) analysis, while Dynamic Features are extracted using a pre-trained VGG-16 model. The Hybrid Feature Space combines static and dynamic features, optimized using the Mean Dropout Feature Selection Technique (MDFS).

4. Model Training: The hybrid feature space is used to train an XGBoost model, evaluated using precision, recall, and F1-score metrics.

The proposed model demonstrated high performance in multi-class classification of lung conditions, achieving an accuracy of 95.54% without mean dropout and 96.18% with mean dropout. High scores were achieved for precision, recall, and F1-score across all four classes (bacterial pneumonia, COVID-19, normal lungs, and viral pneumonia). The reported Area Under the Curve (AUC) values were 0.99 for bacterial and viral pneumonia, 0.96 for normal, and 0.95 for COVID-19.

The ESN-MDFS model surpassed existing lightweight models in both accuracy (achieving 96.18% compared to the next best at 95.54%) and compactness (889 KB compared to the next best at 12 MB). The authors demonstrated that the proposed model is ideal for deployment on resource-constrained edge devices, facilitating immediate diagnostic insights and more rapid treatment decisions.

Suggestions:

1. Review the paper for grammatical errors and ensure subject-verb agreement and correct tense usage throughout the manuscript.

2. Ensure that all terms and concepts are clearly defined and consistently used throughout the paper. Consider adding a table of abbreviations to improve comprehensibility and avoid ambiguity.

3. Be consistent when reporting results. Use the same format for all metrics and comparisons. For example, in Table 5, ensure that the number of samples in each category is presented consistently across all discussed models.

4. Outline potential future work, including improvements to the model and adaptation to other imaging modalities.

Reviewer #3: 1. The paper's formatting needs improvement for consistency. Please ensure that the table formatting is uniform throughout the document. Additionally, standardize the captions for figures and tables to maintain a cohesive style. Lastly, align the equations consistently, either centering them or left-aligning them.

2. The comparasion experiments should be run multiple times to show the mean and std of the results.

3. Considering that VGG16 is a well-known model, it may not be necessary to include the entire model architecture as a figure or provide extensive introductions to it. Consider simplifying this section to focus on the more novel aspects of your work.

4. While using a fixed model for feature extraction is a common technique, the concepts of "Dynamic Features" and "Static Features" in this context are unclear. Please provide a clear definition of what you consider dynamic and static features and how they relate to your proposed method.

5. The baseline methods compared in this paper appear to be outdated, with references [61]-[63] being papers published in 2020. Given the rapid advancements in this field, it would be beneficial to compare your work with more recent publications. For example, consider including a comparison with the work by Pratiwi et al. (2021), "Effect of CT-scan image resizing, enhancement and normalization on accuracy of covid-19 detection," which also utilizes VGG16. Providing a justification for not comparing with more recent works would strengthen your paper.

6. Lack of ablation studies. The paper would benefit from the inclusion of ablation studies to demonstrate the impact of each module in the proposed framework.

7. The novelty of the proposed method is not clearly evident. The combination of data augmentation, using a fixed CNN model for feature extraction, and training an SVM or XGBoost classifier is a well-established and widely used technique across various fields. To enhance the contribution of this paper, consider highlighting the specific innovations or improvements your method offers compared to existing approaches. Clearly articulate how your proposed framework advances the state-of-the-art or addresses limitations of previous methods in the context of your specific application.

6. PLOS authors have the option to publish the peer review history of their article (what does this mean?). If published, this will include your full peer review and any attached files.

Reviewer #1: No

Reviewer #2: No

Reviewer #3: No

---

## [Author Response · Author response to Decision Letter 0]

2 Aug 2024

To: Prof. Dr. Joerg Heber

PLOS ONE, Editor-in-Chief

Re: Manuscript submission to PLOS ONE

PONE-D-24-21569

Enhancing Multiclass COVID-19 Prediction with ESN-MDFS: Extreme Smart Network using Mean Dropout Feature

Selection Technique

Date: July 31, 2024

Dear Dr. Heber

We are pleased to inform you that we have revised the manuscript in the light of reviewers’ comments. The reviewers’ recommendations were extremely useful, and we have addressed all their recommendations in the revised manuscript. Please see below for responses to each individual comment. We hope that the revisions in the manuscript and our accompanying responses will be sufficient to make our manuscript suitable for publication in PLOS ONE.

S. No. Comments Rebuttal

Reviewer 1 Comments

1 This research, by virtue of the fact, stress the existence and severity of dire results associated with delayed and improper diagnosis which can be a real threat to the survival of humankind. The latest studies have revealed that deep learning models can be used for coronavirus image classification. - add the following work for this statement such as: A Deep learning-based X-Ray Imaging Diagnosis System for Classification of Tuberculosis, COVID19, and Pneumonia Traits using Evolutionary Algorithm The authors [15] proposed a novel automated framework for the classification of tuberculosis, COVID-19, and pneumonia from chest x-ray images using deep learning and an improved optimization technique. The proposed deep learning-based framework achieved high classification accuracy 98.2%, 99.0%, and 98.7%) on three different datasets for tuberculosis, COVID-19, and pneumonia detection from chest X-ray images. The authors employed the Wilcoxon signed-rank test to statistically validate the superior performance of their proposed method. The integration of feature fusion was instrumental in enhancing the method's accuracy.

2 While deep learning models have shown promise in classifying COVID-19 images, there are still limitations that need to be addressed. One such limitation is the lack of research on applying domain adaptation techniques to overcome the

challenge of the cross-dataset problem. - add the following work for this statement such as: Deep learning network selection and optimized information fusion for enhanced COVID-19 detection The researchers [16] proposed a wrapper-based technique to improve the classification performance of chest infection (including COVID-19) detection using X-rays by extracting deep features using pretrained deep learning models and optimizing them using various optimization techniques, while also using a network selection technique to select the deep learning models. The proposed deep learning framework achieved a high classification accuracy of 97.7% in detecting chest infections, including COVID-19. Rigorous validation confirmed the framework's reliability for classifying both COVID-19 and other chest infections, suggesting its potential as a valuable tool for clinicians.

3 Related work of this manuscript can be further enhanced by adding few recent works. Also, discuss the cutting edges

and gaps that should be linked with the proposed work. The issues have been addressed. Some related work from recent studies as suggested by the reviewers have been incorporated in the literature and discussion sections. Moreover, future directions also been added.

4 How many features are extracted from the VGG model? How many layers used for the re-training on the selected

datasets? The VGG16 model was employed to extract 1024 features. To adapt the model to the specific task, the final four layers were fine-tuned using the selected datasets

5 What is the purpose of GLCM features and how many features are extracted? The Gray Level Co-occurrence Matrix (GLCM) is a statistical technique used to extract texture features from images by analyzing the spatial relationships between pixel intensities. Its applications span various domains, including SAR imagery for land cover classification (water, vegetation, urban areas) [48] and medical imaging for detecting retinal abnormalities [49], where color features have shown superior accuracy. GLCM calculates the relative frequency of pixel pairs with specific intensities and spatial configurations, creating a matrix from which statistical measures can be derived. For this study, we computed 25 GLCM features, subsequently reducing them to 17 using MeanDropout feature selection.

6 What is the loss function of the XGBoost classifier and what is the purpose of this classifier? XGBoost traditionally uses convex loss functions, recent research has explored custom and non-convex loss functions to enhance performance in specific applications [59]. For instance, [60] investigated the use of squared logistics loss (SqLL) to improve accuracy. [59] developed weighted softmax loss functions for industrial applications, while [61] proposed a generalized XGBoost method accommodating both convex and some non-convex loss functions. These advancements demonstrate XGBoost's versatility and potential for tailored solutions in various domains, including big data analysis and multi-objective parameter regularization.

The purpose of the XGBoost classifier is multifaceted and versatile, as evidenced by various research studies. XGBoost is utilized for enhancing prediction accuracy in diverse fields such as meteorology for hailstorm forecasting [62], detecting patterns in financial datasets to differentiate between solvable and bankrupt situations [63], improving learner performance prediction in Intelligent Tutoring Systems by enhancing models like Performance Factor Analysis and DAS3H [64], and detecting malware in Internet of Medical Things (IoMT) data for better medical assistance through dimensionality reduction and efficient classification [65]. The XGBoost algorithm's scalability, robustness, and proficiency with complex datasets make it a valuable tool for increasing prediction accuracy, addressing class imbalances, enhancing performance prediction models, and improving data analysis in various domains.

Reviewer 2 Comments

1 Review the paper for grammatical errors and ensure subject-verb agreement and correct tense usage throughout the

manuscript. The issue has been addressed

2 Ensure that all terms and concepts are clearly defined and consistently used throughout the paper. Consider adding a

table of abbreviations to improve comprehensibility and avoid ambiguity. The issue has been addressed and highlighted in red color

3 Be consistent when reporting results. Use the same format for all metrics and comparisons. For example, in Table 5,

ensure that the number of samples in each category is presented consistently across all discussed models. The issue has been addressed and highlighted in red color

4 Outline potential future work, including improvements to the model and adaptation to other imaging modalities. The issue has been addressed and highlighted in red color in the introduction section

Reviewer 3 Comments

1 The paper's formatting needs improvement for consistency. Please ensure that the table formatting is

uniform throughout the document. Additionally, standardize the captions for figures and tables to maintain a cohesive

style. Lastly, align the equations consistently, either centering them or left aligning them. The issue has been addressed

2 The comparison experiments should be run multiple times to show the mean and std of the results. 

Fig. 10. Fold vs accuracy curve to distinguish multi-class using ESN-MDFS

Figure 10 illustrates the accuracy of the ESN-MDFS model across seven cross-validation folds. The model achieved a mean accuracy of 95.57% with a standard deviation of 0.54, demonstrating consistent performance across different data subsets.

3 Considering that VGG16 is a well-known model, it may not be necessary to include the entire model architecture as a

figure or provide extensive introductions to it. Consider simplifying this section to focus on the more novel aspects of

your work. The issue has been addressed and details have been incorporated according as advised by the esteemed reviewer

4 While using a fixed model for feature extraction is a common technique, the concepts of "Dynamic Features" and

"Static Features" in this context are unclear. Please provide a clear definition of what you consider dynamic and static

features and how they relate to your proposed method. Deep features extracted from VGG-19 provide a powerful representation of image content. They capture high-level semantic information about the image, such as the presence of specific objects or patterns. In the context of COVID-19 classification, these features can effectively discriminate between different lung pathologies, including pneumonia, viral pneumonia, and COVID-19. By leveraging the hierarchical structure of VGG-19, these features can capture subtle visual patterns that are often challenging for traditional image processing techniques.

Static GLCM features, on the other hand, provide complementary information about the texture and spatial relationships between pixels in an image. These features are sensitive to image patterns and structures, which can be crucial for differentiating between different types of lung abnormalities. By combining deep features and GLCM features, it is possible to create a more robust and discriminative feature space for multi-class COVID-19 classification.

5 The baseline methods compared in this paper appear to be outdated, with references [61]-[63] being papers

published in 2020. Given the rapid advancements in this field, it would be beneficial to compare your work with more recent publications. For example, consider including a comparison with the work by Pratiwi et al. (2021), "Effect of CT scan image resizing, enhancement and normalization on accuracy of covid-19 detection," which also utilizes VGG16.

Providing a justification for not comparing with more recent works would strengthen your paper. The primary outcome measured in this study [73] is the accuracy of COVID-19 detection using CT-scan images and various preprocessing methods. The main findings of this study are that different preprocessing methods, including resizing, enhancement, and normalization, had an impact on the accuracy of COVID-19 classification using a deep learning model (VGG-16), and the highest accuracy of 88.54% was achieved using a combination of deformed resizing, CLAHE enhancement, and normalization to the range of [0 1] and [-1 1].

Initial model comparisons were conducted using contemporary models available at the time of results generation. However, we acknowledge the value of the reviewers' suggestions to benchmark against the most recent state-of-the-art research.

6 Lack of ablation studies. The paper would benefit from the inclusion of ablation studies to demonstrate the impact of each module in the proposed framework. As per suggestion we calculate the results through VGG16 as a single module. 

a) b)

c) d)

Fig. 5 Multi-class Covid-19 detection using VGG-16, a) Confusion Matrix, b) Classification Report, c) AUC, d) Accuracy-Loss Curve

Figure 5 presents the multi-class COVID-19 detection results exclusively based on VGG-16 deep features, as visualized through confusion matrices (5a), classification reports (5b), AUC curves (5c), and accuracy loss curves (5d). Relying solely on deep features, the model achieved an overall accuracy of 93% in classifying the four target classes. Notably, the AUC for multi-class differentiation (bacterial, COVID-19, viral) was 0.99, while perfect discrimination (AUC of 1.00) was observed for the normal class.

As per suggested, we obtain the results through XGBoost also, the results are presented follow: -

a) b)

c)

Fig. 6 Multi-class Covid-19 detection using XGBoost, a) Confusion Matrix, b) Classification Report, c) AUC, d) Accuracy-Loss Curve

Figure 6 illustrates the multi-class COVID-19 detection performance solely based on XGBoost-processed static features, as depicted in the confusion matrix (6a), classification report (6b), and AUC curve (6c). Relying exclusively on static features, the model achieved an overall accuracy of 86% in classifying the four target classes.

7 The novelty of the proposed method is not clearly evident. The combination of data augmentation, using a fixed CNN

model for feature extraction, and training an SVM or XGBoost classifier is a well-established and widely used technique

across various fields. To enhance the contribution of this paper, consider highlighting the specific innovations or

improvements your method offers compared to existing approaches. Clearly articulate how your proposed framework

advances the state-of-the-art or addresses limitations of previous methods in the context of your specific application. This study enhances multiclass COVID-19 prediction through a novel approach encompassing the following key elements:

• Optimized pre-processing: Chest X-ray image quality was improved using techniques such as interpolation, data cleaning, augmentation, feature engineering, image enhancement, morphological operations, segmentation, and transformation.

• Feature extraction: Dynamic VGG-19 and static GLCM features were computed from multiclass data to capture diverse image characteristics.

• Feature selection: A hybrid feature space (HFS) was refined using feature selection methods to eliminate redundant features, thereby improving prediction performance and model size for efficient deployment on edge devices

• The optimal HFS was then utilized to the robust optimized XGBoost algorithm for improved prediction

• Hyperparameter tuning: The hyperparameters of the XGBoost machine learning algorithm were meticulously optimized.

Deep features extracted from VGG-19 provide a powerful representation of image content. They capture high-level semantic information about the image, such as the presence of specific objects or patterns. In the context of COVID-19 classification, these features can effectively discriminate between different lung pathologies, including pneumonia, viral pneumonia, and COVID-19. By leveraging the hierarchical structure of VGG-19, these features can capture subtle visual patterns that are often challenging for traditional image processing techniques.

Static GLCM features, on the other hand, provide complementary information about the texture and spatial relationships between pixels in an image. These features are sensitive to image patterns and structures, which can be crucial for differentiating between different types of lung abnormalities. By combining deep features and GLCM features, it is possible to create a more robust and discriminative feature space for multi-class COVID-19 classification.

The hybrid feature space (HFS)

• Deep features and GLCM features capture different aspects of image information, leading to improved classification performance.

• The combination of these features can better differentiate between subtle visual patterns associated with different lung diseases.

• The use of multiple feature types can help to reduce the impact of noise and variations in image quality.

By effectively fusing these features and employing appropriate machine learning techniques, we developed highly accurate and reliable COVID-19 classification models.

Looking forward to hearing from you.

Regards,

Dr. Lal Hussain

---

## [Editor Report · Decision Letter 1]

7 Aug 2024

PONE-D-24-21569R1Enhancing Multiclass COVID-19 Prediction with ESN-MDFS: Extreme Smart Network using Mean Dropout Feature Selection TechniquePLOS ONE

Dear Dr. Hussain,

Thank you for submitting your manuscript to PLOS ONE. After careful consideration, we feel that it has merit but does not fully meet PLOS ONE’s publication criteria as it currently stands. Therefore, we invite you to submit a revised version of the manuscript that addresses the points raised during the review process.

You did not respond to comment no. 5 from Reviewer 1. Please address it, but in a separate paragraph. Your previous responses were appended to the original comment, making them difficult to follow. Please submit your revised manuscript by Sep 21 2024 11:59PM. If you will need more time than this to complete your revisions, please reply to this message or contact the journal office at plosone@plos.org. Please include the following items when submitting your revised manuscript:A rebuttal letter that responds to each point raised by the academic editor and reviewer(s). You should upload this letter as a separate file labeled 'Response to Reviewers'.A marked-up copy of your manuscript that highlights changes made to the original version. You should upload this as a separate file labeled 'Revised Manuscript with Track Changes'.An unmarked version of your revised paper without tracked changes. You should upload this as a separate file labeled 'Manuscript'.If applicable, we recommend that you deposit your laboratory protocols in protocols.io to enhance the reproducibility of your results. Protocols.io assigns your protocol its own identifier (DOI) so that it can be cited independently in the future. For instructions see: https://journals.plos.org/plosone/s/submission-guidelines#loc-laboratory-protocols. Additionally, PLOS ONE offers an option for publishing peer-reviewed Lab Protocol articles, which describe protocols hosted on protocols.io. Read more information on sharing protocols at https://plos.org/protocols?utm_medium=editorial-email&utm_source=authorletters&utm_campaign=protocols.

We look forward to receiving your revised manuscript.

Kind regards,

Catalin Buiu

Academic Editor

PLOS ONE

Journal Requirements:

Additional Editor Comments:

You did not respond to comment no. 5 from Reviewer 1. Please address it, but in a separate paragraph. Your previous responses were appended to the original comments, making them difficult to follow.

---

## [Author Response · Author response to Decision Letter 1]

8 Aug 2024

To: Prof. Dr. Joerg Heber

PLOS ONE, Editor-in-Chief

Re: Manuscript submission to PLOS ONE

PONE-D-24-21569R1

Enhancing Multiclass COVID-19 Prediction with ESN-MDFS: Extreme Smart Network using Mean Dropout Feature

Selection Technique

Date: August 08, 2024

Dear Dr. Heber

We are pleased to inform you that we have revised the manuscript in the light of reviewers’ comments. The reviewers’ recommendations were extremely useful, and we have addressed all their recommendations in the revised manuscript. Please see below for responses to each individual comment. We hope that the revisions in the manuscript and our accompanying responses will be sufficient to make our manuscript suitable for publication in PLOS ONE.

Journal Requirements:

Response: We have checked each reference and ensured that there is retracted article found.

Additional Editor Comments:

You did not respond to comment no. 5 from Reviewer 1. Please address it, but in a separate paragraph. Your previous responses were appended to the original comments, making them difficult to follow.

Response: We have addressed the comments accordingly.

Reviewers’ comments

S. No. Comments Rebuttal

Reviewer 1 Comments

1 This research, by virtue of the fact, stress the existence and severity of dire results associated with delayed and improper diagnosis which can be a real threat to the survival of humankind. The latest studies have revealed that deep learning models can be used for coronavirus image classification. - add the following work for this statement such as: A Deep learning-based X-Ray Imaging Diagnosis System for Classification of Tuberculosis, COVID19, and Pneumonia Traits using Evolutionary Algorithm The authors [15] proposed a novel automated framework for the classification of tuberculosis, COVID-19, and pneumonia from chest x-ray images using deep learning and an improved optimization technique. The proposed deep learning-based framework achieved high classification accuracy 98.2%, 99.0%, and 98.7%) on three different datasets for tuberculosis, COVID-19, and pneumonia detection from chest X-ray images. The authors employed the Wilcoxon signed-rank test to statistically validate the superior performance of their proposed method. The integration of feature fusion was instrumental in enhancing the method's accuracy.

2 While deep learning models have shown promise in classifying COVID-19 images, there are still limitations that need to be addressed. One such limitation is the lack of research on applying domain adaptation techniques to overcome the

challenge of the cross-dataset problem. - add the following work for this statement such as: Deep learning network selection and optimized information fusion for enhanced COVID-19 detection The researchers [16] proposed a wrapper-based technique to improve the classification performance of chest infection (including COVID-19) detection using X-rays by extracting deep features using pretrained deep learning models and optimizing them using various optimization techniques, while also using a network selection technique to select the deep learning models. The proposed deep learning framework achieved a high classification accuracy of 97.7% in detecting chest infections, including COVID-19. Rigorous validation confirmed the framework's reliability for classifying both COVID-19 and other chest infections, suggesting its potential as a valuable tool for clinicians.

3 Related work of this manuscript can be further enhanced by adding few recent works. Also, discuss the cutting edges

and gaps that should be linked with the proposed work. The issues have been addressed. Some related work from recent studies as suggested by the reviewers have been incorporated in the literature and discussion sections. Moreover, future directions also been added.

4 How many features are extracted from the VGG model? How many layers used for the re-training on the selected

datasets? The VGG16 model was employed to extract 1024 features. To adapt the model to the specific task, the final four layers were fine-tuned using the selected datasets

5 What is the purpose of GLCM features and how many features are extracted? The Gray Level Co-occurrence Matrix (GLCM) is a statistical technique used to extract texture features from images by analyzing the spatial relationships between pixel intensities. Its applications span various domains, including SAR imagery for land cover classification (water, vegetation, urban areas) [48] and medical imaging for detecting retinal abnormalities [49], where color features have shown superior accuracy. Gray Level Co-occurrence Matrix (GLCM) analysis computes the frequency of pixel pairs with specific intensity values and spatial relationships, forming a matrix from which statistical features can be extracted. In this study, 25 GLCM features were initially calculated and subsequently reduced to 17 through MeanDropout feature selection.

.

6 What is the loss function of the XGBoost classifier and what is the purpose of this classifier? XGBoost traditionally uses convex loss functions, recent research has explored custom and non-convex loss functions to enhance performance in specific applications [59]. For instance, [60] investigated the use of squared logistics loss (SqLL) to improve accuracy. [59] developed weighted softmax loss functions for industrial applications, while [61] proposed a generalized XGBoost method accommodating both convex and some non-convex loss functions. These advancements demonstrate XGBoost's versatility and potential for tailored solutions in various domains, including big data analysis and multi-objective parameter regularization.

The purpose of the XGBoost classifier is multifaceted and versatile, as evidenced by various research studies. XGBoost is utilized for enhancing prediction accuracy in diverse fields such as meteorology for hailstorm forecasting [62], detecting patterns in financial datasets to differentiate between solvable and bankrupt situations [63], improving learner performance prediction in Intelligent Tutoring Systems by enhancing models like Performance Factor Analysis and DAS3H [64], and detecting malware in Internet of Medical Things (IoMT) data for better medical assistance through dimensionality reduction and efficient classification [65]. The XGBoost algorithm's scalability, robustness, and proficiency with complex datasets make it a valuable tool for increasing prediction accuracy, addressing class imbalances, enhancing performance prediction models, and improving data analysis in various domains.

Reviewer 2 Comments

1 Review the paper for grammatical errors and ensure subject-verb agreement and correct tense usage throughout the

manuscript. The issue has been addressed

2 Ensure that all terms and concepts are clearly defined and consistently used throughout the paper. Consider adding a

table of abbreviations to improve comprehensibility and avoid ambiguity. The issue has been addressed and highlighted in red color

3 Be consistent when reporting results. Use the same format for all metrics and comparisons. For example, in Table 5,

ensure that the number of samples in each category is presented consistently across all discussed models. The issue has been addressed and highlighted in red color

4 Outline potential future work, including improvements to the model and adaptation to other imaging modalities. The issue has been addressed and highlighted in red color in the introduction section

Reviewer 3 Comments

1 The paper's formatting needs improvement for consistency. Please ensure that the table formatting is

uniform throughout the document. Additionally, standardize the captions for figures and tables to maintain a cohesive

style. Lastly, align the equations consistently, either centering them or left aligning them. The issue has been addressed

2 The comparison experiments should be run multiple times to show the mean and std of the results. 

Fig. 10. Fold vs accuracy curve to distinguish multi-class using ESN-MDFS

Figure 10 illustrates the accuracy of the ESN-MDFS model across seven cross-validation folds. The model achieved a mean accuracy of 95.57% with a standard deviation of 0.54, demonstrating consistent performance across different data subsets.

3 Considering that VGG16 is a well-known model, it may not be necessary to include the entire model architecture as a

figure or provide extensive introductions to it. Consider simplifying this section to focus on the more novel aspects of

your work. The issue has been addressed and details have been incorporated according as advised by the esteemed reviewer

4 While using a fixed model for feature extraction is a common technique, the concepts of "Dynamic Features" and

"Static Features" in this context are unclear. Please provide a clear definition of what you consider dynamic and static

features and how they relate to your proposed method. Deep features extracted from VGG-19 provide a powerful representation of image content. They capture high-level semantic information about the image, such as the presence of specific objects or patterns. In the context of COVID-19 classification, these features can effectively discriminate between different lung pathologies, including pneumonia, viral pneumonia, and COVID-19. By leveraging the hierarchical structure of VGG-19, these features can capture subtle visual patterns that are often challenging for traditional image processing techniques.

Static GLCM features, on the other hand, provide complementary information about the texture and spatial relationships between pixels in an image. These features are sensitive to image patterns and structures, which can be crucial for differentiating between different types of lung abnormalities. By combining deep features and GLCM features, it is possible to create a more robust and discriminative feature space for multi-class COVID-19 classification.

5 The baseline methods compared in this paper appear to be outdated, with references [61]-[63] being papers

published in 2020. Given the rapid advancements in this field, it would be beneficial to compare your work with more recent publications. For example, consider including a comparison with the work by Pratiwi et al. (2021), "Effect of CT scan image resizing, enhancement and normalization on accuracy of covid-19 detection," which also utilizes VGG16.

Providing a justification for not comparing with more recent works would strengthen your paper. The primary outcome measured in this study [73] is the accuracy of COVID-19 detection using CT-scan images and various preprocessing methods. The main findings of this study are that different preprocessing methods, including resizing, enhancement, and normalization, had an impact on the accuracy of COVID-19 classification using a deep learning model (VGG-16), and the highest accuracy of 88.54% was achieved using a combination of deformed resizing, CLAHE enhancement, and normalization to the range of [0 1] and [-1 1].

Initial model comparisons were conducted using contemporary models available at the time of results generation. However, we acknowledge the value of the reviewers' suggestions to benchmark against the most recent state-of-the-art research.

6 Lack of ablation studies. The paper would benefit from the inclusion of ablation studies to demonstrate the impact of each module in the proposed framework. As per suggestion we calculate the results through VGG16 as a single module. 

a) b)

c) d)

Fig. 5 Multi-class Covid-19 detection using VGG-16, a) Confusion Matrix, b) Classification Report, c) AUC, d) Accuracy-Loss Curve

Figure 5 presents the multi-class COVID-19 detection results exclusively based on VGG-16 deep features, as visualized through confusion matrices (5a), classification reports (5b), AUC curves (5c), and accuracy loss curves (5d). Relying solely on deep features, the model achieved an overall accuracy of 93% in classifying the four target classes. Notably, the AUC for multi-class differentiation (bacterial, COVID-19, viral) was 0.99, while perfect discrimination (AUC of 1.00) was observed for the normal class.

As per suggested, we obtain the results through XGBoost also, the results are presented follow: -

a) b)

c)

Fig. 6 Multi-class Covid-19 detection using XGBoost, a) Confusion Matrix, b) Classification Report, c) AUC, d) Accuracy-Loss Curve

Figure 6 illustrates the multi-class COVID-19 detection performance solely based on XGBoost-processed static features, as depicted in the confusion matrix (6a), classification report (6b), and AUC curve (6c). Relying exclusively on static features, the model achieved an overall accuracy of 86% in classifying the four target classes.

7 The novelty of the proposed method is not clearly evident. The combination of data augmentation, using a fixed CNN

model for feature extraction, and training an SVM or XGBoost classifier is a well-established and widely used technique

across various fields. To enhance the contribution of this paper, consider highlighting the specific innovations or

improvements your method offers compared to existing approaches. Clearly articulate how your proposed framework

advances the state-of-the-art or addresses limitations of previous methods in the context of your specific application. This study enhances multiclass COVID-19 prediction through a novel approach encompassing the following key elements:

• Optimized pre-processing: Chest X-ray image quality was improved using techniques such as interpolation, data cleaning, augmentation, feature engineering, image enhancement, morphological operations, segmentation, and transformation.

• Feature extraction: Dynamic VGG-19 and static GLCM features were computed from multiclass data to capture diverse image characteristics.

• Feature selection: A hybrid feature space (HFS) was refined using feature selection methods to eliminate redundant features, thereby improving prediction performance and model size for efficient deployment on edge devices

• The optimal HFS was then utilized to the robust optimized XGBoost algorithm for improved prediction

• Hyperparameter tuning: The hyperparameters of the XGBoost machine learning algorithm were meticulously optimized.

Deep features extracted from VGG-19 provide a powerful representation of image content. They capture high-level semantic information about the image, such as the presence of specific objects or patterns. In the context of COVID-19 classification, these features can effectively discriminate between different lung pathologies, including pneumonia, viral pneumonia, and COVID-19. By leveraging the hierarchical structure of VGG-19, these features can capture subtle visual patterns that are often challenging for traditional image processing techniques.

Static GLCM features, on the other hand, provide complementary information about the texture and spatial relationships between pixels in an image. These features are sensitive to image patterns and structures, which can be crucial for differentiating between different types of lung abnormalities. By combining deep features and GLCM features, it is possible to create a more robust and discriminative feature space for multi-class COVID-19 classification.

The hybrid feature space (HFS)

• Deep features and GLCM features capture different aspects of image information, leading to improved classification performance.

• The combination of these features can better differentiate between subtle visual patterns associated with different lung diseases.

• The use of multiple feature types can help to reduce the impact of noise and variations in image quality.

By effectively fusing these features and employing a

---

## [Editor Report · Decision Letter 2]

14 Aug 2024

PONE-D-24-21569R2Enhancing Multiclass COVID-19 Prediction with ESN-MDFS: Extreme Smart Network using Mean Dropout Feature Selection TechniquePLOS ONE

Dear Dr. Hussain,

Thank you for submitting your manuscript to PLOS ONE. After careful consideration, we feel that it has merit but does not fully meet PLOS ONE’s publication criteria as it currently stands. Therefore, we invite you to submit a revised version of the manuscript that addresses the points raised during the review process.

We look forward to receiving your revised manuscript.

Kind regards,

Catalin Buiu

Academic Editor

PLOS ONE

Journal Requirements:

**Additional Editor Comments:**

You did not respond to the first reviewer's fifth comment:

5) how many parameters are trained for the VGG model and how the hyperparameters are selected? add in the manuscript.

---

## [Author Response · Author response to Decision Letter 2]

16 Aug 2024

To: Prof. Dr. Joerg Heber

PLOS ONE, Editor-in-Chief

Re: Manuscript submission to PLOS ONE

PONE-D-24-21569R1

Enhancing Multiclass COVID-19 Prediction with ESN-MDFS: Extreme Smart Network using Mean Dropout Feature

Selection Technique

Date: August 08, 2024

Dear Dr. Heber

We are pleased to inform you that we have revised the manuscript in the light of reviewers’ comments. The reviewers’ recommendations were extremely useful, and we have addressed all their recommendations in the revised manuscript. Please see below for responses to each individual comment. We hope that the revisions in the manuscript and our accompanying responses will be sufficient to make our manuscript suitable for publication in PLOS ONE.

Journal Requirements:

Response: We have checked each reference and ensured that there is retracted article found.

Additional Editor Comments:

You did not respond to comment no. 5 from Reviewer 1. Please address it, but in a separate paragraph. Your previous responses were appended to the original comments, making them difficult to follow.

Response: We have addressed the comments accordingly.

Reviewers’ comments

S. No. Comments Rebuttal

Reviewer 1 Comments

1 This research, by virtue of the fact, stress the existence and severity of dire results associated with delayed and improper diagnosis which can be a real threat to the survival of humankind. The latest studies have revealed that deep learning models can be used for coronavirus image classification. - add the following work for this statement such as: A Deep learning-based X-Ray Imaging Diagnosis System for Classification of Tuberculosis, COVID19, and Pneumonia Traits using Evolutionary Algorithm The authors [15] proposed a novel automated framework for the classification of tuberculosis, COVID-19, and pneumonia from chest x-ray images using deep learning and an improved optimization technique. The proposed deep learning-based framework achieved high classification accuracy 98.2%, 99.0%, and 98.7%) on three different datasets for tuberculosis, COVID-19, and pneumonia detection from chest X-ray images. The authors employed the Wilcoxon signed-rank test to statistically validate the superior performance of their proposed method. The integration of feature fusion was instrumental in enhancing the method's accuracy.

2 While deep learning models have shown promise in classifying COVID-19 images, there are still limitations that need to be addressed. One such limitation is the lack of research on applying domain adaptation techniques to overcome the

challenge of the cross-dataset problem. - add the following work for this statement such as: Deep learning network selection and optimized information fusion for enhanced COVID-19 detection The researchers [16] proposed a wrapper-based technique to improve the classification performance of chest infection (including COVID-19) detection using X-rays by extracting deep features using pretrained deep learning models and optimizing them using various optimization techniques, while also using a network selection technique to select the deep learning models. The proposed deep learning framework achieved a high classification accuracy of 97.7% in detecting chest infections, including COVID-19. Rigorous validation confirmed the framework's reliability for classifying both COVID-19 and other chest infections, suggesting its potential as a valuable tool for clinicians.

3 Related work of this manuscript can be further enhanced by adding few recent works. Also, discuss the cutting edges

and gaps that should be linked with the proposed work. The issues have been addressed. Some related work from recent studies as suggested by the reviewers have been incorporated in the literature and discussion sections. Moreover, future directions also been added.

4 How many features are extracted from the VGG model? How many layers used for the re-training on the selected datasets? The VGG16 model was employed for feature extraction, generating 1024-dimensional feature vectors for each image in the dataset. To adapt the model to the specific characteristics of our target problem, the final four fully connected layers of VGG16 were re-trained on the selected datasets.

5 How many parameters are trained for the VGG model and how the hyperparameters are selected? add in the manuscript. Total trainable parameters for VGG-16 were 1,051,648 (first Dense) + 2,099,328 (second Dense) + 4,100 (final Dense) = 3,155,076 trainable parameters. To adapt the model to the specific task, the final four layers were fine-tuned using the selected datasets. Optimal performance hinges on careful selection of hyperparameters and we chosen the learning rate, optimizer, batch size, epochs, and regularization techniques by optimizing the hyperparameters using Bayesian optimization to fine-tune these hyperparameters.

6 What is the purpose of GLCM features and how many features are extracted? The Gray Level Co-occurrence Matrix (GLCM) is a statistical technique used to extract texture features from images by analyzing the spatial relationships between pixel intensities. Its applications span various domains, including SAR imagery for land cover classification (water, vegetation, urban areas) [48] and medical imaging for detecting retinal abnormalities [49], where color features have shown superior accuracy. Gray Level Co-occurrence Matrix (GLCM) analysis computes the frequency of pixel pairs with specific intensity values and spatial relationships, forming a matrix from which statistical features can be extracted. In this study, 25 GLCM features were initially calculated and subsequently reduced to 17 through MeanDropout feature selection.

.

7 What is the loss function of the XGBoost classifier and what is the purpose of this classifier? XGBoost traditionally uses convex loss functions, recent research has explored custom and non-convex loss functions to enhance performance in specific applications [59]. For instance, [60] investigated the use of squared logistics loss (SqLL) to improve accuracy. [59] developed weighted softmax loss functions for industrial applications, while [61] proposed a generalized XGBoost method accommodating both convex and some non-convex loss functions. These advancements demonstrate XGBoost's versatility and potential for tailored solutions in various domains, including big data analysis and multi-objective parameter regularization.

The purpose of the XGBoost classifier is multifaceted and versatile, as evidenced by various research studies. XGBoost is utilized for enhancing prediction accuracy in diverse fields such as meteorology for hailstorm forecasting [62], detecting patterns in financial datasets to differentiate between solvable and bankrupt situations [63], improving learner performance prediction in Intelligent Tutoring Systems by enhancing models like Performance Factor Analysis and DAS3H [64], and detecting malware in Internet of Medical Things (IoMT) data for better medical assistance through dimensionality reduction and efficient classification [65]. The XGBoost algorithm's scalability, robustness, and proficiency with complex datasets make it a valuable tool for increasing prediction accuracy, addressing class imbalances, enhancing performance prediction models, and improving data analysis in various domains.

Reviewer 2 Comments

1 Review the paper for grammatical errors and ensure subject-verb agreement and correct tense usage throughout the

manuscript. The issue has been addressed

2 Ensure that all terms and concepts are clearly defined and consistently used throughout the paper. Consider adding a

table of abbreviations to improve comprehensibility and avoid ambiguity. The issue has been addressed and highlighted in red color

3 Be consistent when reporting results. Use the same format for all metrics and comparisons. For example, in Table 5,

ensure that the number of samples in each category is presented consistently across all discussed models. The issue has been addressed and highlighted in red color

4 Outline potential future work, including improvements to the model and adaptation to other imaging modalities. The issue has been addressed and highlighted in red color in the introduction section

Reviewer 3 Comments

1 The paper's formatting needs improvement for consistency. Please ensure that the table formatting is

uniform throughout the document. Additionally, standardize the captions for figures and tables to maintain a cohesive

style. Lastly, align the equations consistently, either centering them or left aligning them. The issue has been addressed

2 The comparison experiments should be run multiple times to show the mean and std of the results. 

Fig. 10. Fold vs accuracy curve to distinguish multi-class using ESN-MDFS

Figure 10 illustrates the accuracy of the ESN-MDFS model across seven cross-validation folds. The model achieved a mean accuracy of 95.57% with a standard deviation of 0.54, demonstrating consistent performance across different data subsets.

3 Considering that VGG16 is a well-known model, it may not be necessary to include the entire model architecture as a

figure or provide extensive introductions to it. Consider simplifying this section to focus on the more novel aspects of

your work. The issue has been addressed and details have been incorporated according as advised by the esteemed reviewer

4 While using a fixed model for feature extraction is a common technique, the concepts of "Dynamic Features" and

"Static Features" in this context are unclear. Please provide a clear definition of what you consider dynamic and static

features and how they relate to your proposed method. Deep features extracted from VGG-19 provide a powerful representation of image content. They capture high-level semantic information about the image, such as the presence of specific objects or patterns. In the context of COVID-19 classification, these features can effectively discriminate between different lung pathologies, including pneumonia, viral pneumonia, and COVID-19. By leveraging the hierarchical structure of VGG-19, these features can capture subtle visual patterns that are often challenging for traditional image processing techniques.

Static GLCM features, on the other hand, provide complementary information about the texture and spatial relationships between pixels in an image. These features are sensitive to image patterns and structures, which can be crucial for differentiating between different types of lung abnormalities. By combining deep features and GLCM features, it is possible to create a more robust and discriminative feature space for multi-class COVID-19 classification.

5 The baseline methods compared in this paper appear to be outdated, with references [61]-[63] being papers

published in 2020. Given the rapid advancements in this field, it would be beneficial to compare your work with more recent publications. For example, consider including a comparison with the work by Pratiwi et al. (2021), "Effect of CT scan image resizing, enhancement and normalization on accuracy of covid-19 detection," which also utilizes VGG16.

Providing a justification for not comparing with more recent works would strengthen your paper. The primary outcome measured in this study [73] is the accuracy of COVID-19 detection using CT-scan images and various preprocessing methods. The main findings of this study are that different preprocessing methods, including resizing, enhancement, and normalization, had an impact on the accuracy of COVID-19 classification using a deep learning model (VGG-16), and the highest accuracy of 88.54% was achieved using a combination of deformed resizing, CLAHE enhancement, and normalization to the range of [0 1] and [-1 1].

Initial model comparisons were conducted using contemporary models available at the time of results generation. However, we acknowledge the value of the reviewers' suggestions to benchmark against the most recent state-of-the-art research.

6 Lack of ablation studies. The paper would benefit from the inclusion of ablation studies to demonstrate the impact of each module in the proposed framework. As per suggestion we calculate the results through VGG16 as a single module. 

a) b)

c) d)

Fig. 5 Multi-class Covid-19 detection using VGG-16, a) Confusion Matrix, b) Classification Report, c) AUC, d) Accuracy-Loss Curve

Figure 5 presents the multi-class COVID-19 detection results exclusively based on VGG-16 deep features, as visualized through confusion matrices (5a), classification reports (5b), AUC curves (5c), and accuracy loss curves (5d). Relying solely on deep features, the model achieved an overall accuracy of 93% in classifying the four target classes. Notably, the AUC for multi-class differentiation (bacterial, COVID-19, viral) was 0.99, while perfect discrimination (AUC of 1.00) was observed for the normal class.

As per suggested, we obtain the results through XGBoost also, the results are presented follow: -

a) b)

c)

Fig. 6 Multi-class Covid-19 detection using XGBoost, a) Confusion Matrix, b) Classification Report, c) AUC, d) Accuracy-Loss Curve

Figure 6 illustrates the multi-class COVID-19 detection performance solely based on XGBoost-processed static features, as depicted in the confusion matrix (6a), classification report (6b), and AUC curve (6c). Relying exclusively on static features, the model achieved an overall accuracy of 86% in classifying the four target classes.

7 The novelty of the proposed method is not clearly evident. The combination of data augmentation, using a fixed CNN

model for feature extraction, and training an SVM or XGBoost classifier is a well-established and widely used technique

across various fields. To enhance the contribution of this paper, consider highlighting the specific innovations or

improvements your method offers compared to existing approaches. Clearly articulate how your proposed framework

advances the state-of-the-art or addresses limitations of previous methods in the context of your specific application. This study enhances multiclass COVID-19 prediction through a novel approach encompassing the following key elements:

• Optimized pre-processing: Chest X-ray image quality was improved using techniques such as interpolation, data cleaning, augmentation, feature engineering, image enhancement, morphological operations, segmentation, and transformation.

• Feature extraction: Dynamic VGG-19 and static GLCM features were computed from multiclass data to capture diverse image characteristics.

• Feature selection: A hybrid feature space (HFS) was refined using feature selection methods to eliminate redundant features, thereby improving prediction performance and model size for efficient deployment on edge devices

• The optimal HFS was then utilized to the robust optimized XGBoost algorithm for improved prediction

• Hyperparameter tuning: The hyperparameters of the XGBoost machine learning algorithm were meticulously optimized.

Deep features extracted from VGG-19 provide a powerful representation of image content. They capture high-level semantic information about the image, such as the presence of specific objects or patterns. In the context of COVID-19 classification, these features can effectively discriminate between different lung pathologies, including pneumonia, viral pneumonia, and COVID-19. By leveraging the hierarchical structure of VGG-19, these features can capture subtle visual patterns that are often challenging for traditional image processing techniques.

Static GLCM features, on the other hand, provide complementary information about the texture and spatial relationships between pixels in an image. These 

---

## [Editor Report · Decision Letter 3]

23 Aug 2024

Enhancing Multiclass COVID-19 Prediction with ESN-MDFS: Extreme Smart Network using Mean Dropout Feature Selection Technique

PONE-D-24-21569R3

Dear Dr. Hussain,

We’re pleased to inform you that your manuscript has been judged scientifically suitable for publication and will be formally accepted for publication once it meets all outstanding technical requirements.

Kind regards,

Catalin Buiu

Academic Editor

PLOS ONE
---

## [Editor Report · Acceptance letter]

11 Sep 2024

PONE-D-24-21569R3 

PLOS ONE

Dear Dr. Hussain, 

I'm pleased to inform you that your manuscript has been deemed suitable for publication in PLOS ONE. Congratulations! Your manuscript is now being handed over to our production team.

Kind regards, 

on behalf of

Dr. Catalin Buiu 

Academic Editor

PLOS ONE